# Genotoxic stress triggers the activation of IRE1α-dependent RNA decay to modulate the DNA damage response

Estefanie Dufey[1,2,3], José Manuel Bravo-San Pedro[4,5], Cristian Eggers[6,7], Matías González-Quiroz [1,2,3,8,9], Hery Urra[1,2,3], Alfredo I. Sagredo[1,2,3], Denisse Sepulveda[1,2,3], Philippe Pihán[1,2,3], Amado Carreras-Sureda[1,2,3], Younis Hazari[1,2,3], Eduardo A. Sagredo[10], Daniela Gutierrez[11], Cristian Valls[11], Alexandra Papaioannou [8,9], Diego Acosta-Alvear[12,13,14], Gisela Campos[15], Pedro M. Domingos [16], Rémy Pedeux[8,9], Eric Chevet [8,9], Alejandra Alvarez[11], Patricio Godoy[15], Peter Walter [12,13], Alvaro Glavic[6,7], Guido Kroemer [4,5,17,18,19] & Claudio Hetz [1,2,3,20✉]

The molecular connections between homeostatic systems that maintain both genome integrity and proteostasis are poorly understood. Here we identify the selective activation of the unfolded protein response transducer IRE1α under genotoxic stress to modulate repair programs and sustain cell survival. DNA damage engages IRE1α signaling in the absence of an endoplasmic reticulum (ER) stress signature, leading to the exclusive activation of regulated IRE1α-dependent decay (RIDD) without activating its canonical output mediated by the transcription factor XBP1. IRE1α endoribonuclease activity controls the stability of mRNAs involved in the DNA damage response, impacting DNA repair, cell cycle arrest and apoptosis. The activation of the c-Abl kinase by DNA damage triggers the oligomerization of IRE1α to catalyze RIDD. The protective role of IRE1α under genotoxic stress is conserved in fly and mouse. Altogether, our results uncover an important intersection between the molecular pathways that sustain genome stability and proteostasis.

[1] Biomedical Neuroscience Institute (BNI), Faculty of Medicine, University of Chile, Santiago, Chile. [2] Center for Geroscience, Brain Health and Metabolism (GERO), Santiago, Chile. [3] Program of Cellular and Molecular Biology, Institute of Biomedical Sciences, University of Chile, Santiago, Chile. [4] Centre de Recherche des Cordeliers, Equipe labellisée par la Ligue contre le cancer, Inserm U1138, Université de Paris, Sorbonne Université, Paris, France. [5] Metabolomics and Cell Biology Platforms, Gustave Roussy Cancer Campus, Villejuif, France. [6] Department of Biology, Faculty of Sciences, University of Chile, Santiago, Chile. [7] Center for Genome Regulation, Faculty of Sciences, University of Chile, Santiago, Chile. [8] Proteostasis & Cancer Team, INSERM U1242, University of Rennes 1, Rennes, France. [9] Centre de Lutte contre le Cancer Eugène Marquis, Rennes, France. [10] Department of Molecular Biosciences, The Wenner-Gren Institute, Stockholm University, Svante Arrheniusväg 20C, 106 91 Stockholm, Sweden. [11] Department of Cell & Molecular Biology, Pontificia Universidad Católica de Chile, 8331010 Santiago, Chile. [12] Department of Biochemistry and Biophysics, University of California, San Francisco, CA, USA. [13] Howard Hughes Medical Institute, University of California, San Francisco, San Francisco, CA, USA. [14] Department of Molecular, Cellular, and Developmental Biology, University of California, Santa Barbara, Santa Barbara, CA 93106, USA. [15] IfADo-Leibniz Research Centre for Working Environment and Human Factors at the Technical University Dortmund, 44139 Dortmund, Germany. [16] Instituto de Tecnologia Química e Biológica, Universidade Nova de Lisboa, Av. da República, 2780-157 Oeiras, Portugal. [17] Pôle de Biologie, Hôpital Européen Georges Pompidou, AP-HP, Paris, France. [18] Suzhou Institute for Systems Medicine, Chinese Academy of Medical Sciences, Suzhou, China. [19] Department of Women's and Children's Health, Karolinska Institute, Karolinska University Hospital, Stockholm, Sweden. [20] The Buck Institute for Research in Aging, Novato, CA 94945, USA. ✉email: chetz@med.uchile.cl

The integrity of the genome is constantly threatened by endogenously produced toxic metabolites, physical, and chemical insults, resulting in a variety of DNA lesions. Inefficient DNA repair translates into cellular dysfunction and death, but also into the propagation of somatic mutations and malignant transformation. To limit genome instability, cells engage the DNA damage response (DDR) and activate repair mechanisms to reverse or minimize alterations in DNA integrity[1]. The DDR pathway involves the interconnection of complex signaling networks that enforce cell cycle arrest and DNA repair. The failure of this adaptive mechanism is detrimental for the cell, resulting in an irreversible cell cycle arrest (senescence) or the activation of different types of regulated death programs[1]. Accordingly, perturbations in the DDR largely contribute to oncogenesis, tumor progression, and the resistance to irradiation and chemotherapy with genotoxic drugs. The accumulation of synonymous mutations, aneuploidy, as well as the activation of oncogenes, deregulate proteostasis[2]. The endoplasmic reticulum (ER) is the main subcellular compartment involved in protein folding and quality control[3], representing a central node of the proteostasis network. The unfolded protein response (UPR) is a specialized mechanism to cope with ER stress[4,5], that also influences most hallmarks of cancer[6]. Nevertheless, the possible involvement of the UPR in the surveillance and maintenance of genome integrity remains elusive.

Inositol requiring enzyme 1 alpha (known as ERN1, referred to as IRE1α hereafter) controls the most evolutionary conserved UPR signaling branch, regulating ER proteostasis and cell survival through distinct functional outputs[4]. IRE1α is a serine/threonine protein kinase and endoribonuclease that catalyzes the unconventional splicing of the mRNA encoding X-Box binding protein 1 (Xbp1), generating an active transcription factor that enforces adaptive programs[7]. IRE1α also degrades a subset of mRNAs and microRNAs through a process known as regulated IRE1α-dependent decay of RNA (RIDD), impacting various biological processes, including cell death and inflammation[8–11]. A screen aiming to define the universe of XBP1-target genes under ER stress identified a cluster of DDR-related components[12], and suboptimal DNA repair may trigger ER stress[2]. Together, these observations suggest a link between DNA damage and ER proteostasis. Here we investigate the possible contribution of IRE1α to the DDR. Surprisingly, we observed that genotoxic stress engages IRE1α signaling in the absence of ER stress markers. In fibroblasts undergoing DNA damage, IRE1α activation results in the selective activation of RIDD in the absence of XBP1 mRNA splicing, impacting genome stability, cell survival and cell cycle control. At the molecular level, we identify specific RIDD mRNA substrates as possible effectors of the phenotypes triggered by IRE1α deficiency. We also validated the significance of IRE1α signaling to the DDR in vivo using genetic manipulation in mouse and fly models. Our results suggest that IRE1α has an alternative function in cells undergoing genotoxic stress, where it serves to amplify and sustain an efficient DDR to maintain genome stability and cell survival.

## Results

### DNA damage selectively induces IRE1α signaling toward RIDD.
Upon ER stress, IRE1α dimerization leads to its autotransphosphorylation and the formation of large clusters that are needed for optimal signaling[13]. Exposure of mouse embryonic fibroblasts (MEF) to the DNA damaging agent etoposide, a topoisomerase II inhibitor, triggers mild IRE1α phosphorylation (Fig. 1a) and formation of IRE1α clusters, as revealed using an IRE1α-GFP reporter (Fig. 1b). Similar results were obtained in cells exposed to γ-irradiation (Supplementary Fig. 1a). Unexpectedly, MEF cells

stimulated with etoposide or γ-irradiation failed to engage Xbp1 mRNA splicing, as determined by two independent PCR-based assays (Fig. 1c, d) or western blot analysis (Supplementary Fig. 1b). Moreover, no signs of ER stress were observed in cells undergoing DNA damage when we assessed canonical markers of UPR activation, including the expression of CHOP, ATF4, BiP, as well as ATF6 processing and the phosphorylation of both PERK and eIF2α (Supplementary Fig. 1c, d). As positive controls of DNA damage, we monitored the levels of phosphorylation of the histone H2AX (γ-H2AX) or the upregulation of the cyclin-dependent kinase inhibitor CDKN1A (also known as p21) (Supplementary Fig. 1b, d). Unexpectedly, classical RIDD mRNAs substrates such as Bloc1s1 and Sparc[8,9] decayed upon exposure to DNA damaging agents (Fig. 1e). Importantly, this decrease in Bloc1s1 and Sparc mRNAs did not occur in IRE1α-deficient cells (Fig. 1e), nor upon pharmacological inhibition of the RNase activity of IRE1α with MKC-8866 (Supplementary Fig. 1e, f), confirming the occurrence of RIDD. These results suggest that DNA damage selectively stimulates IRE1α activity toward RIDD and not Xbp1 mRNA splicing in the absence of global ER stress markers.

### IRE1α regulates DDR signaling under genotoxic stress.
To evaluate the significance of IRE1α expression to the adaptive capacity of cells undergoing DNA damage, we compared the viability of IRE1α-deficient and control (wild type, WT) cells after exposure to various agents that induce distinct types of DNA lesions, including etoposide, 5-hydroxyurea, 5-fluorouracil, and γ-irradiation. Remarkably, IRE1α deficiency sensitized cells to all types of genotoxic stress, thus increasing the incidence of cell death (Fig. 1f and Supplementary Fig. 2a). We confirmed these results by measuring caspase-3 activation, a marker of apoptosis (Fig. 1g). We then stably reconstituted IRE1α null cells with an HA-tagged version of IRE1α (IRE1α-HA) that expresses levels similar to endogenous (described in refs. [14,15]). Importantly, the hypersensitivity of IRE1α deficient cells to DNA damage was partially reverted by expressing IRE1α-HA, suggesting that the phenotypes observed in IRE1α-deficient cells under genotoxic stress are a primary phenotype, and are not due to clonal effects or compensatory changes (Supplementary Fig. 2b). In sharp contrast, IRE1α knockout MEFs did not reveal any differential sensitivity to the ER stress inducer tunicamycin, in line with prior results based on pharmacological inhibition of IRE1α[16]. Also, the pharmacological inhibition of IRE1α RNase activity with MKC-8866 increases the susceptibility to cell death after prolonged genotoxic treatments (Fig. 1h, i). These findings highlight the importance of IRE1α in the adaptation response to DNA damage.

The downstream effects of the DDR are mediated by the activity of check point kinases CHK1 and CHK2, engaging the tumor suppressor protein P53 to induce cell-cycle arrest and the transcription of DNA damage-responsive genes, or to trigger apoptotic cell death[17]. DDR signaling translates into the phosphorylation of histone H2AX (γ-H2AX), and the rate of γ-H2AX decay after DNA injury is a sign of DNA repair. Thus, we monitored the kinetics of H2AX (de)phosphorylation after exposing cells to a pulse of etoposide. IRE1α null cells exhibited a lower response and faster attenuation of γ-H2AX phosphorylation compared to cells in which IRE1α was reintroduced (Fig. 2a). Similar results were obtained when cells where exposed to a shorter pulse of a higher concentration of etoposide (Supplementary Fig. 2c). Analysis of nuclear γ-H2AX foci by immunofluorescence and confocal microscopy also indicated reduced γ-H2AX phosphorylation in IRE1α deficient cells (Fig. 2b). To corroborate these results, we performed the comet assay to directly visualize the DNA damage observing that IRE1α knockout cells exposed to etoposide exhibited increased

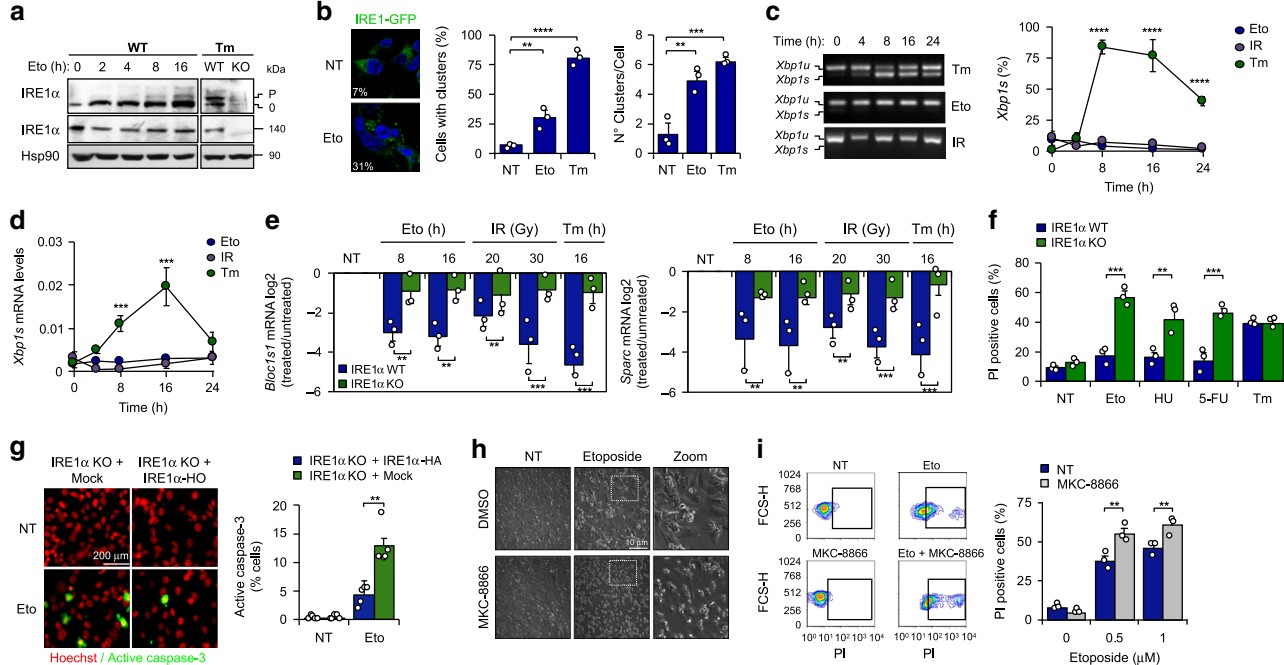

**Fig. 1 Selective activation of RIDD under DNA damage. a** MEF were treated with 10 µM etoposide (Eto) for indicated time points and phosphorylation levels of IRE1α were detected by Phostag assay (p: phosphorylated 0: non-phosphorylated bands). IRE1α levels were analyzed by western blot. Treatment with 500 ng/mL tunicamicyn (Tm) as positive control (8 h) (n = 3). **b** TREX-IRE1-3F6H-GFP cells were treated with 25 µM Eto (8 h) or 1 µg/mL Tm (4 h). IRE1-GFP foci were quantified by confocal microscopy (>200 cells, n = 3). **c** MEF were treated with either 100 ng/mL Tm, 10 µM Eto or 25 Gγ of ionizing radiation (IR) at indicated time points. Xbp1 mRNA splicing percentage was calculated by RT-PCR using densitometric analysis (left panel) (n = 3). **d** Xbp1s mRNA levels were quantified by real-time-PCR in samples described in **c** (n = 3). **e** WT and IRE1α KO cells were treated with 10 µM Eto (8 h and 16 h), IR (20 or 30 Gγ, 16 h) and the decay of mRNA levels of Bloc1s1 and Sparc was monitored by real-time-PCR. Treatment with 500 ng/mL Tm as positive control (n = 3). **f** WT and IRE1α KO cells were treated with 25 µM Eto, 1 µM 5-fluorouracil (5-FU), 1 µM hydroxyurea (HU) or 1 µg/mL Tm by 24 h and viability analyzed using propidium iodide (PI) staining and FACS (n = 3). **g** IRE1α KO (Mock) and IRE1α-HA reconstituted cells were treated with 10 µM Eto for 12 h and apoptosis monitored by caspase-3 positive cells (green). Nucleus (Red) was stained to visualize cells number (n = 3). **h** MEF cells treated 36 h with 0.5 µM Eto in combination with the IRE1α inhibitor 25 µM MKC-8866. Representative images. **i** WT cells were treated with 0.5 and 1 µM Eto or in combination with 25 µM MKC-8866 (36 h). Cell viability analyzed using PI staining and FACS (n = 3). All panels data is shown as mean ± s.e.m.; *p < 0.05, **p < 0.01, and ***p < 0.001, based on **b** two-tailed unpaired t-Student's test, (**c**, **d**). One-way ANOVA followed Tukey's test (**e**–**g**, **i**), two-way ANOVA followed Bonferroni's test. Data is provided as a Source Data file.

DNA damage (Fig. 2c). We also performed a cytokinesis-block micronucleus cytome (CBMN) assay as an indirect measure of DNA breaks. Cells were incubated with etoposide for 3 h, followed by the administration of cytochalasin-B (CytB) for an additional 24 h. IRE1α deficiency resulted in a higher percentage of cells with binucleated nuclei, micronuclei or nuclear buds (Fig. 2d). Altogether, these results suggest that IRE1α null cells have reduced engagement of DNA repair programs.

A direct consequence of DNA damage is cell cycle arrest. No significant differences were observed in cell proliferation (Supplementary Fig. 3a) or cell cycle progression between control and IRE1α null cells in normal conditions. Nevertheless, the fraction of cells arrested in the S and G2/M phases was reduced in IRE1α knockout MEFs exposed to etoposide (Fig. 2e). Similar results were obtained when we reintroduced IRE1α in knockout cells (Supplementary Fig. 3b). At the molecular level, the ablation of IRE1α expression resulted in a strong attenuation in the phosphorylation of CHK1 and CHK2 in DNA-damaged cells (Fig. 2f). In contrast, the phosphorylation of the apical kinase ataxia telangiectasia mutated kinase (ATM; an upstream sensor of the DDR) was not altered in IRE1α knockout cells exposed to etoposide (Fig. 2f and Supplementary Fig. 3c). Consistent with these results, IRE1α knockout and WT MEF cells showed similar phosphorylation of SMC1[18] and KAP1[19], two direct substrates of ATM (Supplementary Fig. 3d), suggesting that the defects

triggered by IRE1α deficiency occur downstream of ATM activation regulating CHK1 and CHK2 activation. Moreover, IRE1α deficient cells showed a sustained kinetic of activation of P53 (Supplementary Fig. 3e). Taken together, these results suggest that IRE1α deficiency deregulates DDR signaling, thus affecting cell cycle progression, DNA repair and cell survival under genotoxic stress.

**Genotoxic stress triggers RIDD to regulate DDR signaling.** To obtain mechanistic insights, we attempted to identify RIDD target mRNAs that might connect IRE1α signaling to the DDR. Previously, a global in vitro screening uncovered a cluster of mRNAs containing consensus sequences cleaved by IRE1α that are associated to a stem-loop structurally similar to the Xbp1 mRNA splicing site[20]. Among the 13 top hits, two DDR-related genes were identified as possible RIDD substrates: PPP2CA-scaffolding A subunit (Ppp2r1a) and RuvB like AAA ATPase1 (Ruvbl1) mRNAs[21] (Fig. 3a). PPP2R1A encodes the scaffold A subunit of protein phosphatase 2 catalytic subunit alpha (PPP2CA, also known as PP2A), which dephosphorylates check point kinases, reversing the G2/M arrest, and directly catalyzing the decay of γ-H2AX phosphorylation and foci[22]. RUVBL1 (also known as Pontin), participates in chromatin remodeling and modulates the stability of DDR protein complexes, thus influencing the dephosphorylation of γ-H2AX[23].

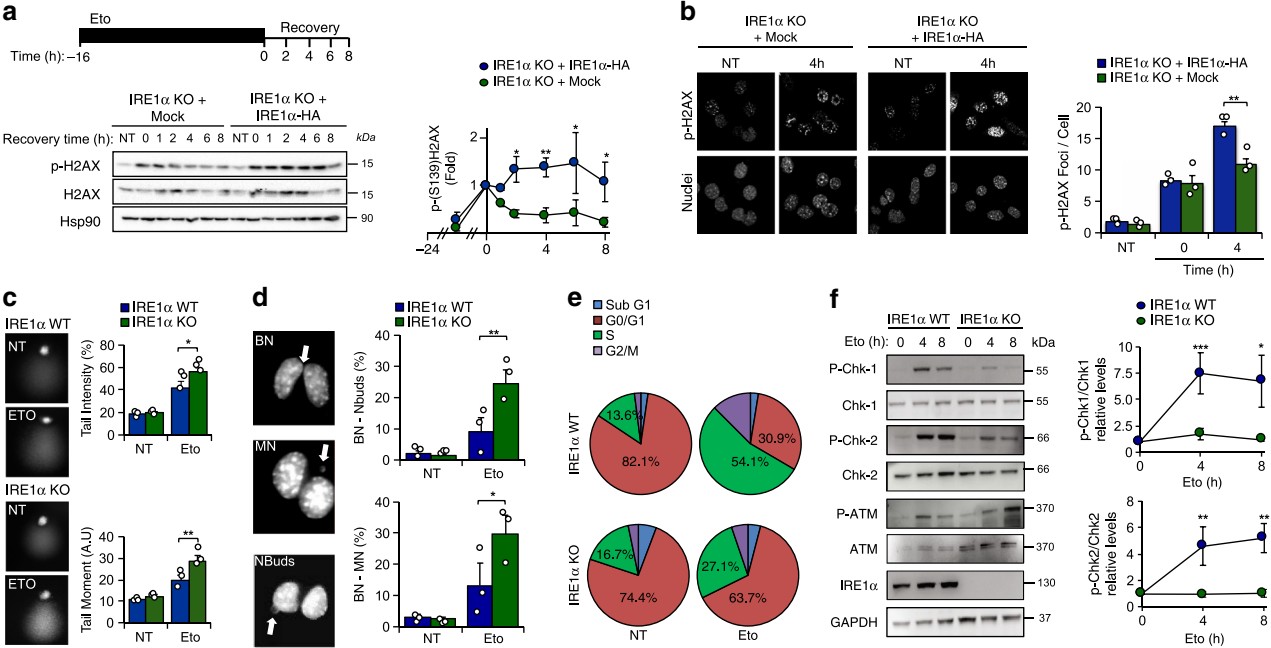

**Fig. 2 IRE1α deficiency impairs the DDR. a** IRE1α KO (Mock) and IRE1α-HA reconstituted cells were pre-incubated with 1 μM etoposide (Eto) for 16 h and washed three times with PBS and fresh cell culture media was added. The decay of phosphorylated H2AX (P-H2AX) was monitored over time by western blot (middle panel). Quantification of the levels of P-H2AX in cells stimulated with Eto (bottom panel) ($n = 3$). **b** IRE1α KO (Mock) and IRE1α-HA reconstituted cells were incubated with 10 μM Eto for 2 h and then washed with PBS and fresh culture media was added. The distribution P-H2AX expression (green) was monitored by indirect immunofluorescence using confocal microscopy. Nuclei were staining with DAPI (Blue). Quantification of P-H2AX per cell is shown (right panel) ($n = 3$). **c** WT and IRE1α KO MEFs cells were treated with 10 μM Eto for 3 h to perform the comet assay. Quantification of tail intensity and tail moment (Tail intensity × tail area) is shown (right panel) ($n = 3$). **d** WT and IRE1α KO MEFs cells were treated with 5 μM Eto for 3 h to determined cytokinesis-block micronucleus cytome assay (CBMC). Binucleated cells (BN) with micronucleus (MN), nuclear buds (Nbuds) or nucleoplasmid bridges (NPB; see arrows) were visualized and quantified using epifluorescence microscopy ($n = 3$). **e** WT and IRE1α KO MEFs cells were treated with 10 μM Eto for 8 h and cell cycle was analyzed by propidium iodide (PI) staining. Quantification of the percentage of cells in G0/G1 and S phases is shown. **f** WT and IRE1α KO MEFs cells were treated with 10 μM Eto for indicated times. Expression and phosphorylation levels of indicated proteins involved in the DDR were monitored by western blot (left panel). Quantification of the levels of p-CHK1 and p-CHK2 is shown (right panel) ($n = 3$). In all panels, data is shown as mean ± s.e.m.; *$p < 0.05$, **$p < 0.01$, and ***$p < 0.001$, based on (**a**, **c**, **d**, **f**) two-way ANOVA followed Bonferroni's test, **b** two-way ANOVA followed Tukey's test. Data is provided as a Source Data file.

Quantification of *Ppp2r1a and Ruvbl1* mRNA levels in cells treated with etoposide demonstrated a decay that was dependent on IRE1α expression (Fig. 3b). These effects on mRNA levels translated into reduced protein expression of PP2A and RUVBL1 only in wild-type cells exposed to etoposide and the basal upregulation in IRE1α null cells (Fig. 3c). In a cell-free assay, recombinant IRE1α directly cleaves a fragment of the Ppp2r1a mRNA that contains the RIDD consensus site (spanning nucleotides 1336-1865), but not an adjacent fragment (Fig. 3d). Similarly, IRE1α exhibited RNase activity on *Ruvbl1* mRNA, thus cleaving this substrate as efficiently as it's known targets *Bloc1s1* mRNA and *Xbp1* mRNA (Fig. 3d). This reaction was suppressed by the IRE1α inhibitor 4μ8C (Fig. 3d).

The lack of *Xbp1* mRNA splicing under DNA damage conditions might involve inhibitory signals, for example mediated by the downregulation of the tRNA ligase RTCB, the targeting of the *Xbp1* mRNA to the ER membrane, or the activity of other regulatory components that are part of IRE1α clusters and component associated with them[24]. Analysis of RTCB levels revealed no changes in IRE1a knockout cells undergoing DNA damage (Supplementary Fig. 4a). To test if DNA damage inhibits *Xbp1* mRNA splicing, we pre-treated cells with tunicamycin for 2 h and then added etoposide at different time points. Remarkably, etoposide failed to interfere with *Xbp1* mRNA splicing induced by tunicamycin (Fig. 3e). Virtually identical results were obtained when a pulse of etoposide was performed followed by the

stimulation of ER stress (Fig. 3g). In contrast, an additive effect was observed on the decay of *Bloc1s1* and *Sparc* mRNAs when ER stress and DNA damaging agents were combined (Fig. 3f, h). These results indicate that DNA damage selectively engages RIDD yet does not cause active suppression of *Xbp1* mRNA splicing.

Considering that PP2A and Pontin are upregulated in IRE1α null cells under genotoxic stress and are involved in the DDR, we attempted to reverse the phenotype of those cells by depleting *Ppp2r1a* or *Ruvbl1* mRNA with suitable short hairpin RNAs (shRNAs) (Supplementary Fig. 4b). Remarkably, knocking down *Ppp2r1a* or *Ruvbl1* in IRE1α null cells augmented the levels of phosphorylated γ-H2AX foci after etoposide treatment during the recovery phase (Fig. 3i). Similar results were obtained when phosphorylated γ-H2AX was monitored by immunoblot (Supplementary Fig. 4c). Moreover, knocking-down *Ppp2r1a* and *Ruvbl1* expression in IRE1α null cells reestablished normal levels of CHK1 phosphorylation (Fig. 3j and Supplementary Fig. 4d, e) and increased population of cells in S/G2M after etoposide treatment (Supplementary Fig. 4f). Taken together, these experiments suggest that the regulation of PP2A and Pontin by IRE1α contributes to DDR signaling under genotoxic treatment.

**c-Abl triggers IRE1α activation under DNA damage.** Recent studies suggest that IRE1α activation can occur independently from ER stress, impacting various biological processes including

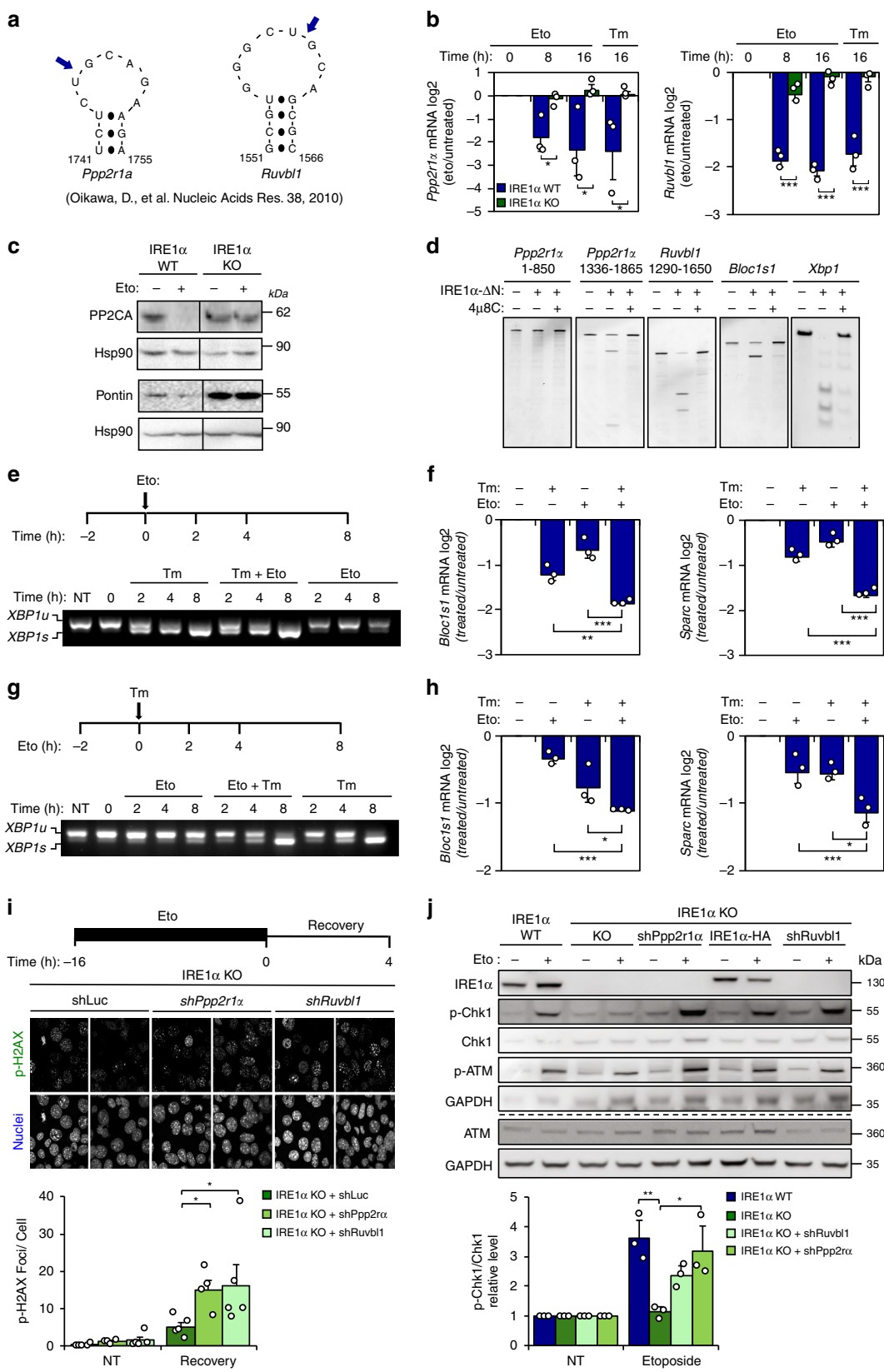

cell migration, synaptic plasticity, angiogenesis and energy metabolism[14,24,25]. However, until now there are no examples for a selective activation of RIDD in the absence of *Xbp1* mRNA splicing. The release of the ER chaperone BiP from the luminal

domain of IRE1α correlates with its activation under ER stress[4]. Co-immunoprecipitation experiments indicated that the BiP-IRE1α interaction decrease after etoposide treatment, but in a lesser extent than ER stress (Supplementary Fig. 5a).

**Fig. 3 IRE1α controls the stability of mRNAs involved in the DRR. a** Putative IRE1α cleavage sites on the *Ppp2r1a* and *Ruvbl1* mRNAs (blue arrows). **b** WT and IRE1α KO MEF cells were treated with 10 µM etoposide (Eto). *Ppp2r1a* and *Ruvbl1* mRNA levels were monitored by real-time-PCR. Treatment with 500 ng/mL tunicamicyn (Tm) as positive control ($n = 3$). **c** Cells were treated with 10 µM Eto (16 h) and PP2A and Pontin expression were monitored by western blot ($n = 3$). **d** In vitro RNA cleavage assay was performed using mRNA fragments of human *Ppp2r1a* and *Ruvbl1*, incubated in the presence or absence of recombinant cytosolic portion of IRE1α (IRE1α-ΔN) protein (30 min). Experiments were performed in presence or absence of IRE1α inhibitor 4µ8C. *Blos1c1* and *Xbp1* mRNA were used as positive controls. **e** Experimental setup (upper panel): MEF cells were pretreated with 100 ng/mL Tm for 2 h and then treated with 10 µM Eto. *Xbp1* mRNA splicing was monitored by RT–PCR (bottom panel). **f** RIDD activity was monitored in samples described in **e** ($n = 3$). **g** Experimental setup (upper panel): MEF WT cells were pretreated with 10 µM Eto for 2 h and then treated with 100 ng/mL Tm. *Xbp1* mRNA splicing was monitored by RT–PCR (bottom panel). **h** RIDD activity was monitored in samples described in **g** ($n = 3$). **i** IRE1α KO MEF cells were transduced with lentiviruses expressing shRNAs against *Ppp2r1a* (shPpp2r1a), *Ruvbl1* (shRuvbl1) or luciferase (shLuc). Cells were incubated with 1 µM Eto (16 h), washed three times with PBS and fresh media was added. P-H2AX levels were monitored by immunofluorescence after 4 h. P-H2AX foci quantification is shown (Bottom panel) (>200 cells, $n = 4$–5). **j** WT, IRE1α KO and reconstituted with IRE1α-HA, expressing *shRuvbl1*, *shPpp2r1a* or *shLuc* cells were treated with 5 µM Eto for 8 h and P-CHK1 and P-ATM monitored by western blot. P-CHK1 quantification is shown (bottom panel) ($n = 3$). All panels data is shown as mean ± s.e.m.; $*p < 0.05$, $**p < 0.01$, and $***p < 0.001$, based on **b** two-way ANOVA followed Bonferroni's test, (**f**, **h–j**) One-way ANOVA followed Tukey's test. Data is provided as a Source Data file.

We then explored possible signaling events that may link the DDR with the induction of RIDD. Interestingly, a recent report indicated that the non-receptor c-Abl tyrosine kinase physically interacts with IRE1α under metabolic stress, allosterically inducing its oligomerization into a conformation that is more likely to catalyze RIDD than *Xbp1* mRNA splicing[26]. Of note, c-Abl has been extensively associated to the DDR, regulating cell-cycle arrest and apoptosis[27]. We confirmed the activation of c-Abl under genotoxic stress (Supplementary Fig. 5b). Treatment of cells with the c-Abl inhibitor imatinib reduced the decay of *Ruvbl1* mRNA in cells exposed to tunicamycin or etoposide (Supplementary Fig. 5c, d). Furthermore, knocking down the expression of c-Abl using shRNAs (Supplementary Fig. 5e) had no effects on the levels of *Xbp1* mRNA splicing (Fig. 4a), but fully prevented the decay of *Ruvbl1* and *Ppp2r1a* mRNAs under ER stress or DNA damage (Fig. 4b), phenocopying the consequences of IRE1α deficiency. Consistent with these results, imatinib treatment attenuated the generation of IRE1α-GFP positive clusters in HEK293 cells undergoing genotoxic stress (Fig. 4c, d). Moreover, we also generated a set of c-Abl null MEF cells using the CRISPR/CAS9 technology (Fig. 4e). The deletion of c-Abl prevented the decay of *Ruvbl1*, *Ppp2r1a* (Fig. 4f) and *Bloc1s1* mRNAs, without an effect on the levels of *Xbp1* mRNA splicing (Supplementary Fig. 5f-g). These observations correlated with the formation of a protein complex between IRE1α and c-Abl as monitored using co-immunoprecipitation in 293T HEK cells overexpressing the proteins (Fig. 4g). We confirmed these results using Proximity Ligation Assay of endogenous c-Abl and IRE1-HA at basal conditions (Fig. 4h) or after exposure to etoposide (Supplementary Fig. 5h, i). Furthermore, using a cell free system, we assessed the effects of recombinant c-Abl on the oligomerization status of purified cytosolic domain of IRE1α. Incubation of purified IRE1α at 37 °C induced its spontaneous oligomerization, which was further enhanced when c-Abl was present in the reaction (Fig. 4i). Taken together, these results suggest that the activation of c-Abl in cells undergoing DNA damage contributes to the selective engagement of RIDD, possibly through a direct interaction with IRE1α.

**IRE1α protect flies against genotoxic stress.** To test the possible role of IRE1α in the maintenance of genome integrity in vivo, we took advantage of *D. melanogaster* as a model organism. The GAL4/UAS system was employed to knockdown the fly orthologue of IRE1 (*dIre1*) using RNAi transgenic animals (Supplementary Fig. 6a). Etoposide treatment failed to trigger an increase in the levels of *dXbp1s* mRNA in larval tissue, as monitored by real time PCR (Fig. 5a). However, larvae fed with etoposide or tunicamycin exhibited a similar reduction in the mRNA levels of *dSparc* and *dMys*, two well-known RIDD targets in flies[28], in addition to *dPontin*, fly orthologue of *Pontin* (Fig. 5b). dIre1 depletion ablated the downregulation of *dSparc*, *dMys* and *dPpp2r1a*, confirming the occurrence of RIDD (Fig. 5b). Moreover, we then determined the impact of dIre1 on the survival of animals under genotoxic conditions and quantified the number of larvae reaching adulthood. Knock-down of dIre1 generated a hypersensitivity phenotype, meaning that most etoposide-treated animals died before reaching maturity (Fig. 5c). Next, we determined the participation of dIre1 in the maintenance of genome integrity. To this end, we performed the somatic mutation and recombination test (SMART). This assay is based on the induction of mutant spots (clones) that arise from loss of heterozygosity in cells of developing animals, which are heterozygous for a recessive wing cell marker mutation (Supplementary Fig. 6b) generating a multiple wing hair (mwh) phenotype (Fig. 5d, left panel). We expressed a dIre1 RNAi construct in the wing imaginal disc using a Nubbin-Gal4 driver (Nub-Gal4). Exposure to doxorubicin increased the number of mutant spots in the fly wing, and this phenotype was exacerbated upon depletion of dIre1 (Fig. 5d, right panel), suggesting compromised DNA repair. Doxorubicin also caused higher rates of apoptosis-associated caspase-3 activation upon dIre1 knockdown (Fig. 5e). Next, we developed a mosaic analysis to ablate dIre1 expression with a repressible cell marker (MARCM), a strategy that allows the comparison of wild-type and mutant cells in the same tissue by assessing GFP expression (Supplementary Fig. 6c). Using this mosaic technology, we generated mutant clones for dIre1 in the eye-antenna imaginal disc and determined the frequency of GFP-positive (dIre1 null cells) and negative cells (WT cells) that persist in the tissue after etoposide treatment. While dIre1 expressing cells maintained their viability after exposure to etoposide (Fig. 5f, left panels), dIre1 null cells proved highly susceptible to this genotoxic agent (Fig. 5f, right panels). Taken together, these results indicate that the fly orthologue of IRE1α protects against genotoxic stress in vivo.

**IRE1α deficiency impairs the DDR in mice.** We then moved forward and investigated the significance of IRE1α expression to the DDR in vivo and deleted the RNase domain of IRE1α in the liver and bone marrow using a conditional knockout (cKO) system controlled by the Mx-Cre system[29]. Poly[I:C] was injected to induce Cre expression, and three weeks later animals were treated with a single dose of either etoposide or tunicamycin, followed by the analysis of liver tissue. A well-established mammalian model of ER stress consists in the intraperitoneal injection

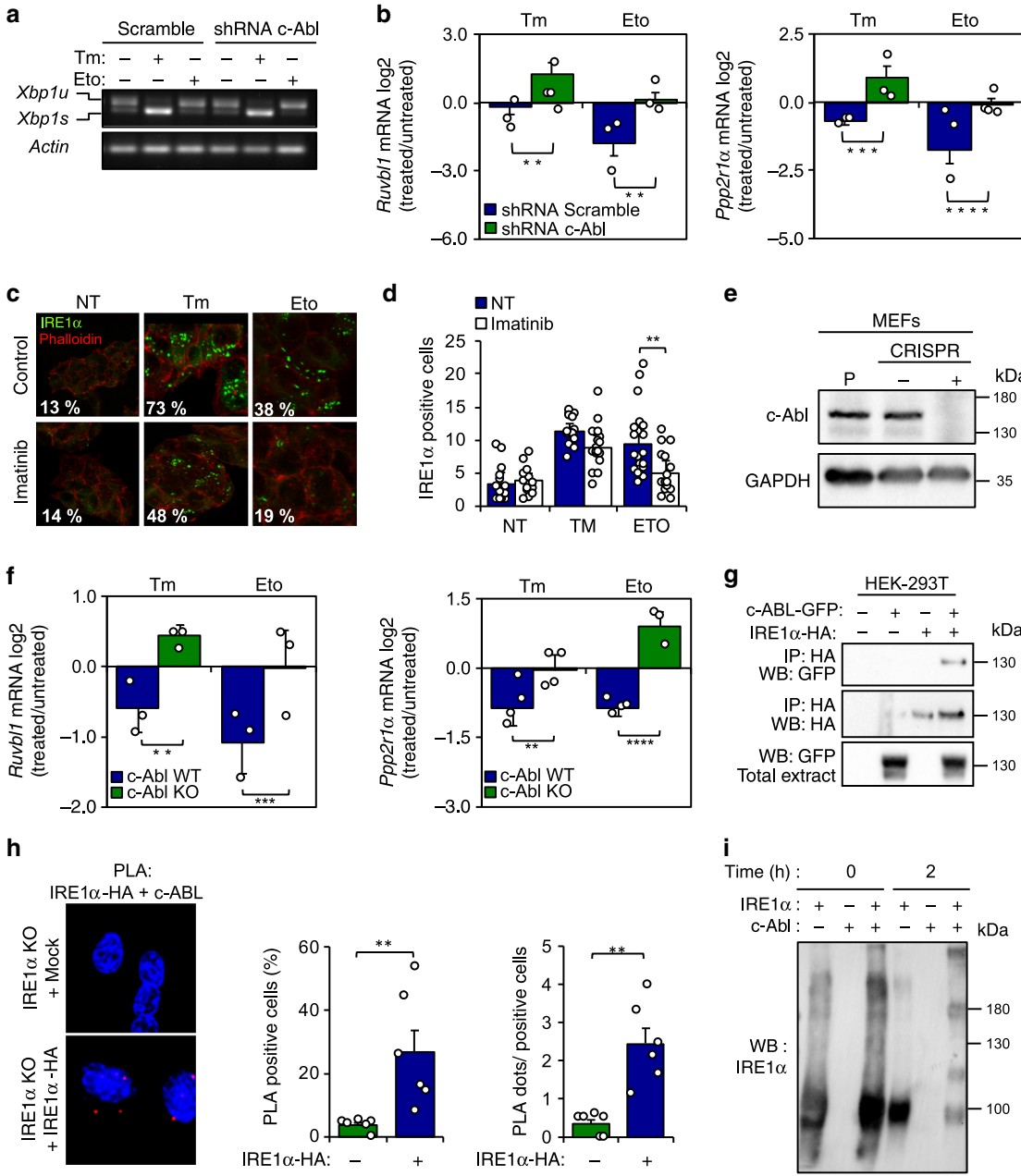

**Fig. 4 c-Abl contributes to the RIDD activation under DNA damage. a** c-Abl was knocked down through the stable delivery of an shRNA. Then cells were treated with 10 μM Etoposide (Eto) or 500 ng/mL tunicamicyn (Tm) for 8 h and *Xbp1* mRNA splicing was monitored by RT–PCR (*n* = 3). **b** MEF ShRNA Scramble and ShRNA c-Abl cells were treated with 10 μM Eto or 500 ng/mL Tm for 8 h, and the decay of *Ppp2r1a* and *Ruvbl1* was measured by real-time-PCR (*n* = 3). **c** Trex-IRE1-GFP cells were pre-treated with 10 μM imatinib by 1 h, and then treated with 10 μM Eto, or 500 ng/mL Tm for 8 h and IRE1-GFP foci visualized by confocal microscopy. **d** Quantification of the percentage of cells positive IRE1-GFP clusters is shown (>200 cells, *n* = 3). **e** c-Abl expression in CRISPR control and c-Abl KO cells was monitored by western blot (*n* = 3). **f** CRISPR control and c-Abl KO cells were treated with 10 μM Eto or 500 ng/mL Tm for 8 h, and the decay of *Ruvbl1* and *Ppp2r1a* was measured by Real-Time-PCR (*n* = 3). **g** HEK-293T cells reconstituted with IRE1α-HA and c-Abl-GFP were exposed to 10 μM Eto for 8 h. Immunoprecipitation (IP) was performed using the HA epitope (IRE1α) and GFP (c-Abl) to assess the possible interaction with c-Abl. **h** IRE1α KO (Mock) and reconstituted cells with an IRE1α-HA were treated 8 h with 10 μM Eto and stained with a proximity ligation assay (PLA) using an anti-HA or anti-c-Abl antibodies and analyzed by confocal microscopy. Right panel: Number of dots per cell analyzed and percentage of PLA positive cells were quantified (*n* = 6). **i** Recombinant IRE1α and c-Abl proteins were incubated at indicated time points and assess its possible interaction by western blot. All data represents the mean ± s.e.m. of three independent experiments, except for co-IP that were performed twice. **p* < 0.05, ***p* < 0.01, and ****p* < 0.001, based on (**b**, **f**) two-way ANOVA followed Bonferroni's test, (**d**, **h**) One-way ANOVA followed Tukey's test. Data is provided as a Source Data file.

of tunicamycin, which elicits a rapid stress response in the liver. Although evident signs of DNA damage were observed in both control and IRE1αcKO animals (indicated by a rise in *p21* mRNA) (Fig. 6a upper panel and Supplementary Fig. 7a), no *Xbp1* mRNA

splicing was detected in the etoposide treated group (Fig. 6a, bottom panel). In sharp contrast, a clear down-regulation of *Ppp2r1a* and *Bloc1s1* mRNA levels occurred in the livers (Fig. 6b) and bone marrows (Supplementary Fig. 7b) from control (but not

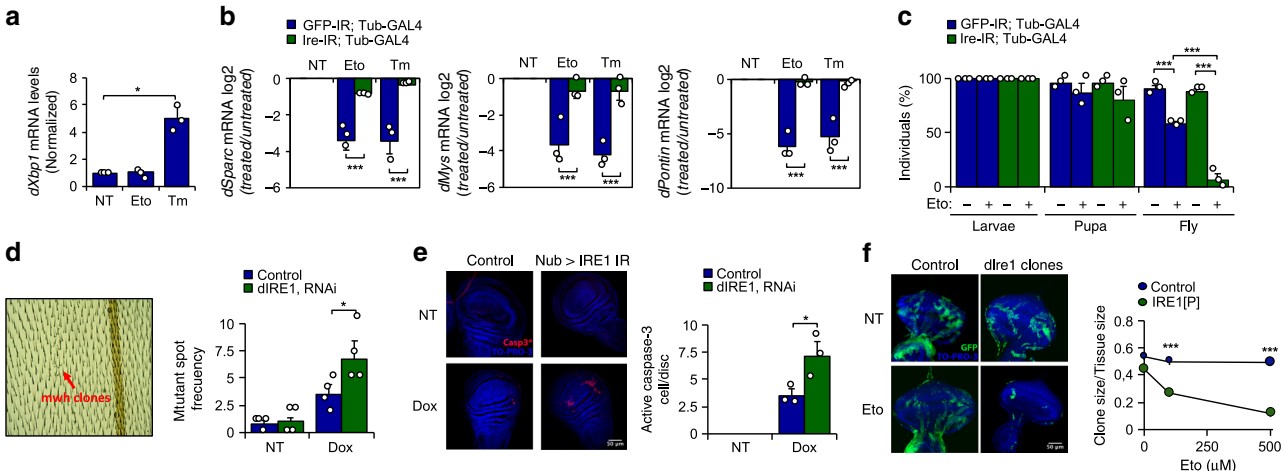

**Fig. 5 IRE1α expression confers protection against genotoxic stress in fly models. a** *D. melanogaster* larvae were fed with 100 μM etoposide (Eto) or 50 μg/ml tunicamycin (Tm) for 4 h and then *dXbp1*s mRNA evaluated by real-time PCR and normalized to the expression levels of *dRpl32* gene (*n* = 3). **b** *dIre1* mRNA was knocked down by expressing a specific RNAi constructs under the control of tubulin-Gal4 driver. *D. melanogaster* larvae were fed with 100 μM Eto or 50 μg/ml Tm for 4 h and the decay of RIDD targets *dSparc*, *dPontin*, and *dMys* mRNA was evaluated by RT-qPCR and normalized to the expression levels of *dRpl32* mRNA (*n* = 3). **c** Control and dIre1 knockdown larvae were fed with 500 μM Eto and allowed to reach adulthood for survival analysis. The number of hatched flies was quantified (*n* = 20 per group) (*n* = 3). **d** A dIre1-RNAi expressing fly line was generated to specifically target dIre1 in the imaginal disc of *D. melanogaster*. The wing SMART assays test was used to monitor genomic alterations after targeting dIre1 in flies. Larvae in first instar were grown in food supplemented with the DNA damaging agent 0.125 mg/ml doxorubicin (Dox) or 0.5% DMSO as control. Adult flies from control and dIre1 RNAi larvae were fixed and the left wing analyzed for the number of mwh clones (right panel) (*n* = 3). **e** Using the same experimental setting described in **d**, imaginal discs were collected, fixed and caspase-3 positive cells detected by immunofluorescence. Nucleus was stained with TO-PRO3 to visualize total number of cells. Quantification of active caspase-3 cells per imaginal disc is presented (right panel) (*n* = 3). **f** Mutant knockout dIre1 cells (dIre1 clone) in the eye-antenna imaginal disc were marked with GFP (see methods). Quantification of the ratio clone size/disc size is presented (right panel) (*n* = 10 clones). In all panels, data is shown as mean ± s.e.m.; *\*p* < 0.05, *\*\*p* < 0.01, and *\*\*\*p* < 0.001, based on **a** one-way ANOVA followed Tukey's test, **b–f** two-way ANOVA followed Bonferroni's test. Data is provided as a Source Data file.

IRE1α-deficient) animals injected with etoposide. Again, no *Xbp1* mRNA splicing was detected in the etoposide treated group, whereas exposure of animals to tunicamycin triggered a very mild response in bone marrow tissue (Supplementary Fig. 7c). In addition, IRE1α deficiency in the liver altered the DDR, reflected in reduced phosphorylation of CHK1 in animals injected with etoposide (Fig. 6c). Importantly, ablation of IRE1α resulted in enhanced susceptibility of liver cells to apoptosis measured as enhanced caspase-3 activation (Fig. 6d).

Finally, to assess the significance of IRE1α to the DDR on an unbiased manner, we performed a gene expression profile analysis of liver tissue derived from mice exposed to etoposide or tunicamycin. Pathway enrichment analysis indicated that IRE1α deficiency attenuated the establishment of a global DDR, delayed the expression of cell cycle arrest genes and activated pro-apoptotic pathways (Fig. 6e and Supplementary Fig. 8a). As expected, under ER stress induced by tunicamycin, IRE1α deficiency in the liver significantly impacted the expression of genes involved in proteostasis control in the secretory pathway (trafficking, folding and quality control) (Supplementary Fig. 8b, c). Taken together, these findings demonstrate that IRE1α signaling contributes to maintaining the stability of the genome when cells face DNA damage.

## Discussion

The current study supports a conserved function for IRE1α as a signaling module of the DDR that differs from its canonical role as an UPR mediator. We propose that IRE1α is part of a key decision-making node in a complex interplay between cell survival and DNA repair upon genotoxic stress. In this context, IRE1α regulates the levels of PP2A and RUVBL1 through the selective engagement of RIDD, controlling the kinetics and

amplitude of γ-H2AX phosphorylation. The contribution of IRE1α to genome stability is conserved in evolution from insects to mammals and impacts whole animal survival as demonstrated using flies. Our results suggest a regulatory mechanism in which the RNase domain of IRE1α is selectively regulated to specifically engage RIDD, presumably upon interaction with c-Abl (Fig. 4g, h). This view is consistent with recent studies that connected using unbiased approaches the pathways involved in maintenance of genome integrity and proteostasis, showing that dysregulation of the DDR resulted in protein aggregation and autophagy induction[30,31]. Moreover, previous work demonstrated that the function and structure of the ER is drastically affected by DNA damaging agents used in chemotherapy[32,33]. Other recent reports suggested that chronic ER stress suppresses DNA repair and sensitizes cancer cells to ionizing radiation and chemotherapy[34–37], in addition to enhancing oxidative damage to the DNA[38]. Interestingly, a recent study also reported that XBP1u, the protein encoded by the unsliced version of *Xbp1* mRNA, regulates the stability of TP53, suggesting alternative connections between the UPR and the DDR under resting conditions[39]. Our results suggest that IRE1α specifically affects signaling events regulating the DDR, and not the DNA damage sensing process. IRE1α operates as an amplification loop, impacting the sustained activation of CHK1/2 and the phosphorylation of γ-H2X through the control of the RIDD targets *Ppp2r1a* and *Ruvbl1*, leading to cell cycle arrest and improved DNA repair and as a consequence maintenance of cell survival (see working model in Fig. 6f).

Although RIDD is proposed to be necessary for the maintenance of ER homeostasis[8,10] and to contribute to the pathogenesis of diabetes[5], cancer[40,41], and inflammatory conditions[42–44], most of the available evidence is difficult to interpret due to the concomitant existence of *Xbp1* mRNA splicing. Our study supports a fundamental biological function for RIDD in the maintenance of

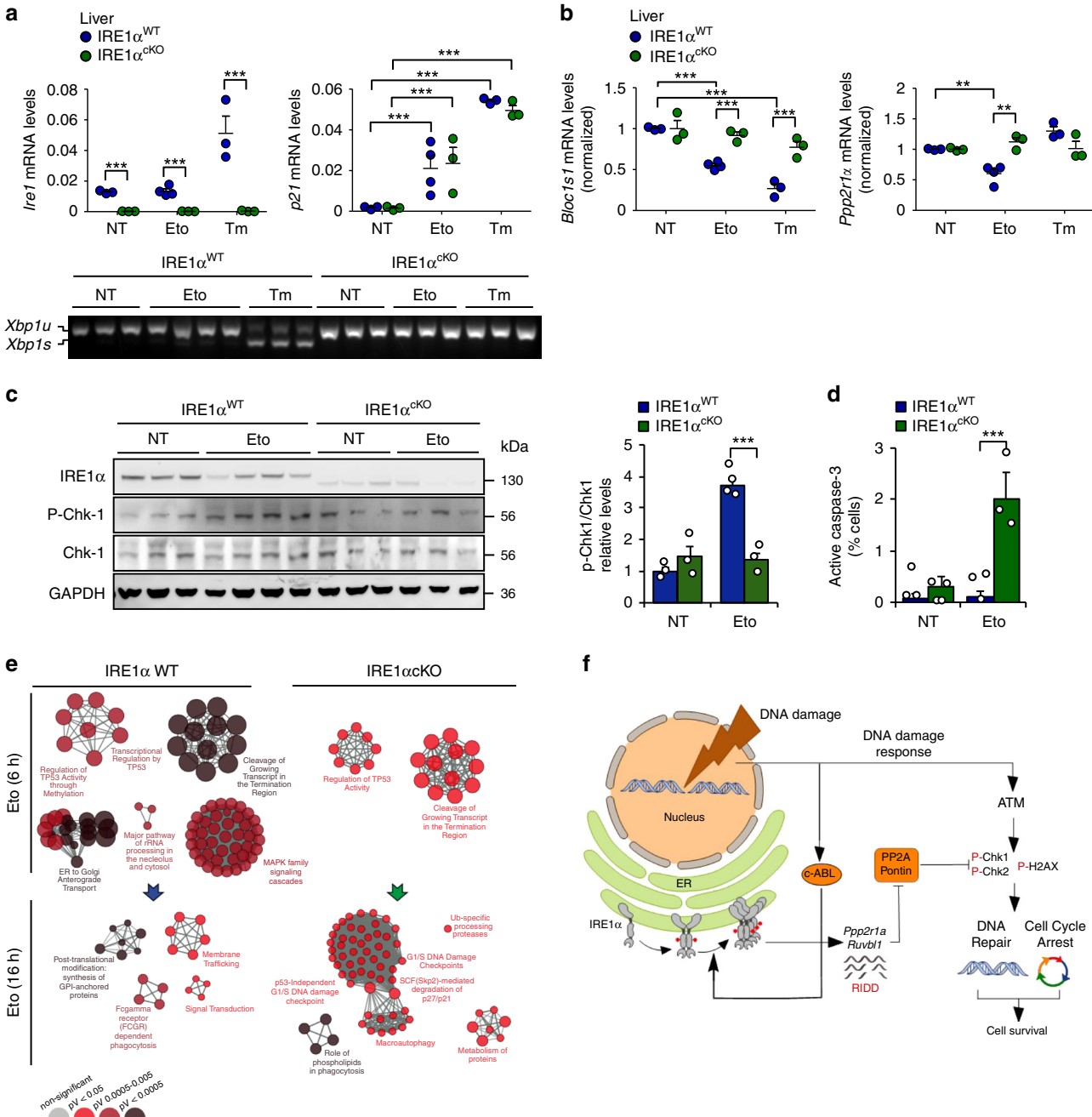

**Fig. 6 IRE1α deletion in liver alters the DDR under genotoxic stress. a** IRE1α was conditionally deleted the liver using the MxCre and LoxP system (IRE1α^cKO). Mice were intraperitoneally injected with 50 mg/Kg etoposide (Eto) or 1 mg/Kg tunicamycin (Tm) and sacrificed 6 h and 16 h later. Total mRNA levels of the deleted *IRE1*, and *p21* were measured 6 h later in the liver by real-time-PCR (*n* = 3–4 mice per group). *Xbp1* mRNA splicing (bottom panel) was monitored in the same samples by RT-PCR. **b** Liver extracts of animals described in **a**, *Ppp2r1a* and *Bloc1s1* mRNA expression levels were measured 6 h later of Eto treatment by real-time-PCR (*n* = 3). **c** Protein liver extracts were obtained from mice treated described in **a** and the expression levels of indicated proteins were monitored 6 h later of Eto treatment by western blot. Quantification of the levels of p-CHK1 is shown (Right panel). **d** Mice from **a** were intraperitoneally injected with 50 mg/Kg Eto and sacrificed 48 h later. Then, livers active-caspase 3 detected by immunohistochemistry (*n* = 2–3). **e** Gene expression profile analysis was performed in mRNA from liver extracts of animals described in **a**. Most significant pathways altered by Eto administration in WT and IRE1α null livers are shown. Three independent biological samples were used. In all panels, data is shown as mean ± s.e.m.; *p < 0.05, **p < 0.01, and ***p < 0.001, based on **a–d** two-way ANOVA followed Bonferroni's test. Data is provided as a Source Data file. **f** Working model: genotoxic stress activates IRE1α in the absence of ER stress markers, selectively engaging RIDD. IRE1α degrades mRNAs involved in the DNA damage response encoding for *Ppp2r1a* and *Ruvbl1*, regulating the (de)phosphorylation of the histone H2AX and CHK-1/2. The non-canonical activation of IRE1α involves the participation of the c-Abl kinase that is activated by DNA damage response kinases as ATM. The expression and function of IRE1α is essential to promote survival under DNA damage conditions by controlling cell cycle arrest and DNA repair programs.

genome integrity, representing a unique example for a selective and specific activation of RIDD with clear physiological implications. IRE1α is frequently affected by loss-of-function mutations in cancer[2,45], contrasting with the notion that cancer cells require IRE1α to survive in hypoxic conditions[3,6]. Our present results support the idea that the genetic alterations of IRE1α observed in cancer may synergize with oncogenes to promote genomic instability due to inefficient DNA repair. Altogether, we uncovered a previously unanticipated function of a major UPR signal transducer as an integral component of the DDR, revealing an intimate connection between the pathways that assure the integrity of the proteome and the genome.

## Methods

**Reagents**. Etoposide, doxorubicin, 5-fluorouracil, hidroxyurea, imatinib and 4µ8C were purchased from Sigma Aldrich. Tunicamycin was obtained from Calbiochem EMB Bioscience Inc. IRE1 inhibitor MKC-8866 was provided by Dr. John Patterson (Fosun Orinove). Cell culture media, fetal calf serum, and antibiotics were obtained from Life Technologies and Sigma Aldrich. Fluorescent probes and secondary antibodies coupled to fluorescent markers were purchased from Molecular Probes, Invitrogen. All other reagents were obtained from Sigma or the highest grade available.

**Cell culture and DNA constructs**. All MEF and HEK cells used here were maintained in Dulbecco's modified Eagles medium supplemented with 5% fetal bovine serum, non-essential amino acids and grown at 37 °C and 5% CO2. IRE1α deficient cells were previously described[25]. The production of amphotropic retroviruses using the HEK293GPG packing cell line was performed as described in ref. [46]. Retroviral plasmids were transfected using Efectene (Qiagen, Valencia, CA, USA) according to the manufacturer's protocols. IRE1α-HA expressing retroviruses were previously described in the pMSCV-Hygro vector[46], where IRE1α contains two tandem HA sequences at the C-terminal domain and a precision enzyme site before the HA tag.

**RNA isolation, RT-PCR and real time PCR**. Total RNA was prepared from cells and tissues using Trizol (Invitrogen, Carlsbad, CA, USA), and cDNA was synthesized with SuperScript III (Invitrogen) using random primers p(dN)6 (Roche). Quantitative real-time PCR reactions were performed using SYBRgreen fluorescent reagent and/or EvaGreen$^{TM}$ using a Stratagene Mx3000P system (Agilent Technologies, Santa Clara, CA 95051, United States). The relative amounts of mRNAs were calculated from the values of comparative threshold cycle by using *Actin* as a control and *Rpl19* or *Rpl32* for RIDD substrates in mouse or *D. melanogaster* samples, respectively. All methods for *Xbp1* mRNA splicing assay were previously described in ref. [47] using primers described in Supplementary Table 1.

**Immunoprecipitations**. HEK-293T cells reconstituted with IRE1α-HA and c-Abl-GFP and IRE1α deficient MEF cells stably transduced with retroviral expression vectors for IRE1α-HA or empty vector were incubated in presence or absence of tunicamycin (500 ng/mL for 4 h) or etoposide (10 µM for 16 h). Cell lysates were prepared for immunoprecipitations in lysis buffer (0.5% NP-40, 50 mM NaCl, 50 mM Tris pH 7.6, 50 mM NaF, 1 mM Na3VO4 and protease inhibitors). Immunoprecipitations (IP) were performed as described[48]. In brief, to IP HA-tagged IRE1α, protein extracts were incubated with anti-HA antibody-agarose complexes (Roche), for 4 h at 4 °C, and then washed three times with 1 mL of lysis buffer and then one time in lysis buffer with 350 mM NaCl. Beads were dried and resuspended in Sample Buffer 2x. Samples were heated for 5 min at 95 °C and resolved by SDS-PAGE 8% followed by western blot analysis.

**Western blot analysis**. Cells were collected and homogenized in RIPA buffer (20 mM Tris pH 8.0, 150 mM NaCl, 0.1% SDS, 0.5% Triton X-100) containing a protease inhibitor cocktail (Roche, Basel, Switzerland) in presence of 50 mM NaF and 1 mM Na3VO4. Protein concentration was determined in all experiments by micro-BCA assay (Pierce, Rockford, IL), and 20–40 µg of total protein was loaded onto 8–12% SDS-PAGE minigels (Bio-Rad Laboratories, Hercules, CA) prior transfer onto PVDF membranes. To evaluate IRE1α phosphorylation, SDS-PAGE minigel were made in presence of the 5 µM of Phostag reagent and 10 µM MnCl2. Membranes were blocked using PBS, 0.1% Tween-20 (PBST) containing 5% milk for 60 min at room temperature then probed overnight with primary antibodies. The following antibodies diluted in blocking solution were used in this study: anti-BiP 1:1000 (Abcam, Cat. ab21685); Anti-phosphorylated S139-H2AX 1:5000 (Millipore, Cat. 05-636), anti-DDIT3 (CHOP) 1:1000 (Santa Cruz, Cat. sc-575), anti-XBP1s 1:1000 (Santa Cruz, Cat. 7160), anti-ATF4 1:2000 (Santa Cruz, Cat. sc-200), anti-Hsp90 1:5000 (Santa Cruz, Cat. sc-13119), anti-IRE1α (14C10) (Santa Cruz, Cat. 3294), anti-Chk1 1:1000 (Santa Cruz, Cat. sc-8408), anti-Chk2 1:1000 (Santa Cruz, Cat. sc-17747), anti-phosphorylated-Chk2, Thr68 1:1000 (Santa Cruz, Cat. sc-16297-R), anti-ATM 1:1000 (Santa Cruz, Cat. sc-7129), anti-HA 1:1000

(Roche, Cat. 11666606001), anti-PP2A A 1:1000 (Cell Signaling technology, Cat. 2041S), anti-RuvbL1 1:1000 (Cell Signaling technology, Cat. 12300), anti-eIF2α 1:000 (Cell Signaling technology, Cat. 9722), anti-phosphorylated-eIF2α 1:1000 (Cell Signaling technology, Cat. 9721), anti-PERK 1:1000 (Cell Signaling technology, Cat. 3192), anti-IRE1α 1:1000 (Cell Signaling technology, Cat. 3294), anti-phosphorylated-Chk1, Ser348 1:1000 (Cell Signaling, Cat. 2341), anti-HA 1:1000 (Cell Signalling, Cat. 3724), anti-GAPDH 1:1000 (Cell Signaling, Cat. 21185), anti-Abl 1:1000 (Sigma, Cat. A5844), anti-phosphorylated-c-Abl, pTyr412, 1:1000 (Sigma, Cat. C52490), anti-SMC1 1:1000 (Abcam, ab21583), anti-phosphorylated-SMC1,Ser957, 1:1000 (Abcam, ab137871), anti-KAP1, 1:1000 (Abcam, ab190178), anti-phosphorylated-KAP1, Ser824, 1:1000 (Abcam, ab70369), anti-phosphorylated-ATM Ser 198 1:1000 (MERK, Cat. 05-740), anti-p21 (Santa Cruz, Cat. sc-6246), anti-phospho-p53 (Cell Signalling, Cat. 9286 S), anti-p53 (Santa Cruz, Cat. sc-98), anti-p53 (Santa Cruz, Cat. sc-55476). Bound antibodies were detected with peroxidase-coupled secondary antibodies incubated for 1 h at room temperature and the ECL system.

**IRE1α oligomerization assay**. TREX cells expressing IRE1α-3F6HGFP WT were obtained from Dr. Peter Walter at UCSF and were previously described[13]. TREX cells plated and treated with doxycycline (500 ng/mL for 48 h). Cells were treated with Tm (1 µg/mL) or etoposide (25 µM) for different times points and fixed with 4% paraformaldehyde for 30 min. Nuclei were stained with Hoechst dye. Coverslips were mounted with Fluoromount G onto slides and visualized by confocal microscopy (Fluoview FV1000). The number and size of IRE1α foci was quantified using segmentation and particle analysis of Image J software.

**In vitro oligomerization assay**. In all, 0.5 µg of the cytoplasmic domain of GST-tagged IRE1α (Sino Biologicals) and 0.1 µg of His tagged c-Abl (Carna Biosciences) were incubating and mixing for indicated time points, at 37 °C in a heat block (300 rmp). Total reaction was prepared in 100 µL of oligomerization assay buffer (50 mM Tris-HCl pH 7.5, 100 mM NaCl, 5 mM MnCl2, 5 mM MgCl2, 1 mm DTT, 1 mM ATP). The half of the reaction mixture was mixed with NuPAGE LDS sample buffer (Invitrogen) and loaded on 6% denaturing polyacrylamide gel and subsequently analyzed by western blot.

**Indirect Immunofluorescence**. IRE1α-HA, and γ-H2AX proteins were visualized by immunofluorescence. Cells were fixed for 30 min with 4% paraformaldehyde and permeabilized 0.5% NP-40 in PBS containing 0.5% BSA (bovine serum albumin) for 10 min. After blocking for 1 h with 10% FBS in PBS containing 0.5% BSA, cells were subsequently incubated with anti-HA 1:1000 (Invitrogen, Cat. 715500), anti-phosphorylated-Chk1, Ser348 1:1000 (Cell Signaling, Cat. 2341), anti-cleaved caspase 3, Asp175 1: (Cell Signalling, Cat. 9661) or anti-Phosphorylated S139-H2AX 1:5000 (Millipore, Cat. 05-636) antibodies overnight at 4 °C. Cell were washed three times in PBS containing 0.5% BSA, and incubated with Alexa-conjugated secondary antibodies (Molecular Probes) for 1 h at 37 °C. Nuclei were stained with Hoechst dye. Coverslips were mounted with Fluoromount G onto slides and visualized by confocal microscopy (Fluoview FV1000).

**Automated microscopy**. Cells were seeded in 96-well imaging plates (BD Falcon, Sparks, USA) 24 h before stimulation. Cells were treated with the indicated agents. Subsequently, cells were fixed with 4% PFA and counterstained with 10 µM Hoechst 33342. After blocking for 1 h with 10% FBS in PBS containing 0.5% BSA, cells were subsequently incubated with anti-phosphorylated-Chk1, Ser345 1:1000 (Cell Signaling, Cat. 2348) antibody, overnight at 4 °C. Cell were washed three times in PBS, and incubated with Alexa-conjugated secondary antibodies (Molecular Probes) for 1 h at 37 °C. Images were acquired using an ImageXpress Micro XLS Widefield High-Content Analysis System operated by the MetaXpress® Image Acquisition and Analysis Software (Molecular Devices, Sunnyvale, CA, US). Acquisition was performed by means of a 20X PlanApo objective (Nikon, Tokyo, Japan). Minium 9 views fields per well for 96-wells plate were acquired. MetaXpress® was utilised to segment cells into nuclear area (based on Hoechst 33342 signal). Cell-like objects were segmented and divided into cytoplasmic and nuclear regions as previously reported[49].

**Proximity ligation assay (PLA)**. Cells were seeded in 12 mm cover slips. After the indicated treatments, cells were fixed for 20 min at RT with 4% paraformaldehyde and permeabilized 0.5% NP-40 in PBS containing 0.5% BSA (Bovine serum albumin) for 10 min. After blocking for 1 h with 10% FBS in PBS containing 0.5% BSA, cells were incubated with the indicated antibodies: Anti-HA (Cat: 901514, Biolegend or Cat: 9110, Abcam) and anti-Abl 1:1000 (Sigma, Cat. A5844) overnight at 4 °C following by Duolink manufacturer´s instructions (Duolink®, Sigma-Aldrich). Images were acquired by confocal microscopy (Nikon C2 plus) using a 60X oil objective lens stacking the images every 0.5 µm to cover all the image of interest. Stack images were deconvoluted using Huygens and ImageJ. Stack deconvolved images were reduced to one dimension by sumslices function (ImageJ). Colocalization was performed in thresholded and masked images were used to determine Manders/Pearson's index was calculated with ImageJ plugin.

**Comet assay**. The comet assay was performed as previously described[50]. Briefly, agarose-slides were prepared with 1% low-gelling-temperature agarose and $2 \times 10^4$ cells/ml and submerged in lysis solution (1.2 M NaCl, 100 mM Na$_2$-EDTA, 0.1% sodium lauryl sarcosinate, 0.26 M NaOH (pH > 13)) for 18 h at 4 °C in the dark. Then, carefully slides removed and submerged in room temperature (18 − 25 °C) in rinse solution (0.03 M NaOH, 2 mM Na$_2$EDTA (pH ~12.3)) for 20 min. electrophoresis was conducted in the same solution for 25 min at a voltage of 0.6 V/cm. Finally, slides were stained in a solution containing 2.5 µg/ml of propidium iodide in distilled water for 20 min and observed in a epifluorescence microscopy. Images were analyzed using Comet Assay IV software.

**Cytokinesis-block micronucleus assay**. Cytokinesis-block micronucleus (CBMN) assay was performed as previously described[51]. In brief, cells were treated with 5 etoposide (5 µM for 3 h). Then, cells were washed three times with PBS and incubated for 24 h with fresh media with 5 µM of Cytochalasin-B. Cells were fixed, stained with Hoechst. Binucleated cells (BN) with micronucleus (MN), Nuclear buds (Nbuds) or nucleoplasmid bridges (NPB) were detected and quantified using epifluorescence microscopy.

**In vitro RNA cleavage assay**. Bloc1s1 (NM_001487.3), Ppp2r1a (NM_014225) and RuvbL1 (NM_00370) cDNA were obtained from MGC cDNA library (Dharmacon). Long sense and antisense oligonucleotides containing a minimal T7 RNA polymerase promoter (5′-TAATACGACTCACTATAGG-3′) fused upstream of the sequence containing different fragments of the genes Ppp2r1a, RuvbL1 and Bloc1s1 harboring 5′-EcoRI and 3′-BamHI overhangs were annealed and ligated into the cognate restriction sites of pUC19 (Invitrogen, Life Technologies). Oligonucleotides sequence to clone *Ppp2r1a* and *RuvbL1* fragments were described previously[20]. In vitro transcription reactions were performed with T7 RNA polymerase using the HiScribe T7 high-yield RNA synthesis kit (New England Biolabs) following the manufacturer's recommendations. The transcribed RNA were treated for 20 min with DNase and purified by urea-polyacrylamide gel electrophoresis (urea–PAGE). RNAs recovered from gel fragments by the crush-and-soak method, were precipitated with 300 mM NaOAc and 1 volume of isopropanol. No co-precipitants were employed. The precipitated RNA pellets were desalted by two washes with 70% ice-cold ethanol, air-dried and re-suspended in an appropriate volume of either nuclease-free water or RNA resuspension buffer (20 mM HEPES, 100 mM NaCl, 1 mM Mg(OAc)$_2$). The oligonucleotides sequence used are listed in the Supplementary Table 2.

The cytosolic kinase/ribonuclease domain construct of IRE1α (KR43) was expressed and purified as described previously[52]. In vitro transcribed, PAGE-purified, refolded RNAs (50 ng) were incubated with 0.5 µM IRE1α-KR43 for the indicated times in RNA cleavage buffer (20 mM HEPES pH 7.5, 70 mM NaCl, 2 mM Mg(OAc)$_2$, 1 mM TCEP, and 5% glycerol). Stop solution (10 M urea, 0.1% SDS, 1 mM EDTA, 0.05% xylene cyanol, 0.05% bromophenol blue) was added at five-fold excess to stop the reactions followed by heating at 80 °C for 3 min. The denatured samples were then loaded on 6% TBE–urea gels (Invitrogen, Life Technologies) and the gels stained with SYBR Gold nucleic acid stain (Invitrogen, Life Technologies). As a negative control we utilized the IRE1α inhibitor 4µ8C to a final concentration of 5 µM.

**Viability assay**. In all, $2.0 \times 10^4$ cells were seeded in 48-well plate and the maintained by 24 h in DMEM cell culture media supplemented with 5% bovine fetal serum and non-essential amino acids. Genotoxic and ER stress were induced by adding genotoxic and ER stress agents to the cells at different concentrations, and maintained for 24 h. Then, cell viability was monitored using propidium iodide staining and flow cytometry (BD FACS Canto, Biosciences).

**Mouse model**. *Ern1* floxed mice were previously described[53] and crossed with Mx1-cre transgenic mice to generate a conditional KO animal (IRE1α$^{cKO}$). Deletion was induced by injection of polyinosinic-polycytidylic acid (Poly (I:C)) which efficiently delete the floxed gene in the liver and bone marrow[29]. In all, 5–6-weeks-old mice were intraperitoneally (i.p) injected three times with 250 µg of poly(I:C) each time with 2 day intervals to induce the Cre expression. Mice were used for experiments at least 2–3 weeks after the final poly (I:C) injection. DMSO or 50 mg/Kg etoposide or 1 mg/Kg tunicamycin or were i.p. injected and 6 h later mice were sacrificed as reported[54]. The liver and bone marrow were frozen at −80 °C for biochemical analysis and the right major lobe of the liver was placed in a petri dish (on ice). Liver tissue was washed in PBS to remove the blood and then, it was fixed in 4% paraformaldehyde (72 h) for histological analyses. The animals' works and care was in accordance with institutional guidelines. Institutional Committee for Animal Care and Handling, University of Chile (Protocol CICUA-CBA-0833).

**Fly studies**. Flies were kept at 25 °C on standard medium with a 12–12 dark–light cycle. Drug administration protocol for all experiments is as follows: Larvae were grown in standard fly medium until day 3 after egg laying (AEL). Then they were transferred to fly instant medium (Carolina Biological Supply 2700 York Burlington, NC, USA) complemented with the appropriate drug for the different treatments. Larvae were fed with the corresponding drug-supplemented media until dissection or adulthood.

The UAS-Ire1 IR (v39562) line was obtained from the Vienna Drosophila RNAi Center (VDRC). The following lines were obtained from Bloomington Drosophila Stock Center (BDSC): UAS-GFP IR (BL-44412); Ire1 mutant w1118; PBac{WH} Ire1$^{f02170}$/TM6B (BL-18520); mwh$^1$ mutant (BL-549) and flr$^3$/TM3, Ser stock (BL-2371). All fly strains are listed in the Supplementary Table 3.

**SMART assay**. For the SMART assay, larvae fed in media complemented with 0.125 mg/mL Doxorubicin. Wings of the hatched female flies were fixed in ethanol, mounted in ethanol:lactic acid 1:1, and 15 wings per condition were analyzed at ×400 magnification for the occurrence of mutant clones (N = 4)[55].

**Survival curve and biochemical analysis of fly tissue**. Survival analyses were carried out growing 20 experimental or control larvae in 500 µM etoposide and the number of living animals was quantified at different time points. For real time RT-PCR analysis, total RNA was extracted from third instar larvae (same treatment as the survival analysis animals) using Trizol (Invitrogen, Carlsbad, CA, USA), and cDNA was synthesized with SuperScript III (Invitrogen)[56,57].

For immunohistochemistry, larvae were fed with 0.125 mg/mL doxorubicin. Then third instar larvae were dissected and fixed as described previously[58]. Larvae carcasses were incubated with Anti-caspase-3 (1:100, Cell Signaling) overnight in PBT supplemented with 0.5% BSA at 4 °C, washed four times with PBT for 15 min and stained with Alexa-conjugated secondary antibody (1:200, Molecular Probes) and TO-PRO3 (1 µM, Invitrogen). VectaShield (Vector Laboratories, Burlingame, California, USA) was used as a mounting media. The number of active Caspase-3 positive cells was quantified in 10 wing imaginal discs (N = 3). Images were taken using a Zeiss LSM510 confocal microscope and analyzed with ImageJ software.

**Microarray analyses**. Affymetrix gene expression data were pre-processed using Transcriptome Analysis Console (TAC) (v4.0, ThermoFisher Scientific). Custom mouse Brainarray chip definition (v22) was used to further annotate the DE files with Entrez and gene symbol IDs. For further analysis, just gene transcript with FDR ($a = 0.05$) correction and 1.5 > fold change was considered. For pathway enrichment analysis, data obtained from untreated wild type or IRE1α$^{cKO}$ after poly I:C treatment liver mice tissues were used as reference for tunicamycin (Tm) and etoposide (Eto) treatment comparisons, to further input them in ClueGO (v2.3.2) software using Reactome pathway enrichment database (v09.11.2016). In addition, to visualize patterns in the gene expression and pathway enrichment scores for specific ontologies, heatmaps were generated using RStudio (v0.99.489, R 3.4.1) based on KEGG pathway database gene lists. Genes which show change of the ratios higher (or lower) than 1.5-fold in the arrays at any comparison have been considered as up or downregulated and subjected for functional enrichment analysis. The Bioconductor package 'clusterProfiler' was applied to perform functional enrichment analysis using the following repositories: GO (Gene Ontology), KEGG (Kyoto Encyclopedia of Genes and Genomes), and Reactome Pathways. GEO Dataset ID: GSE130952.

**Statistical analysis**. Results were statistically compared using the Ordinary One-way ANOVA and Two-way ANOVA followed by different multiples comparison post-tests (Tukey's Multiple Comparison Test or Bonferroni's Multiple Comparison Test). When pertinent, Student's $t$-test was performed for unpaired or paired groups. In all plots $p$ values are show as indicated: *: $p < 0.05$, **: $p < 0.01$, ***: $p < 0.001$ and ****: $p < 0.0001$ and were considered significant. n.s: non-significant.

**Reporting summary**. Further information on research design is available in the Nature Research Reporting Summary linked to this article.

## Data availability

The source data underlying Figures and Supplementary Figures are provided as a Source Data file. All data is available from the corresponding author upon reasonable request.

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

## Acknowledgements

We thank Dr. Takao Iwawaki for providing IRE1 flox mice. We Thank Dr. David Ron for providing IRE1 null MEFs. We thank Dr. Alexis Rivas for assistance with Cellomics array scan (FONDEQUIP EQM120164). We thank Javiera Ponce for assistance with animal care. ANID/FONDAP program 15150012, Millennium Institute P09-015-F, CONICYT-Brazil 441921/2016-7, FONDEF ID16I10223, FONDEF D11E1007, and FONDECYT 1180186 (C.H.), and Ecos-Conicyt n° C17S02 (C.H.). In addition, we thank the support from the U.S. Air Force Office of Scientific Research FA9550-16-1-0384, and Muscular Dystrophy Association, US Office of Naval Research-Global (ONR-G) N62909-16-1-2003, and the European Commission R&D MSCA-RISE-734749 (INSPIRED). We thank the support from: FONDECYT 11180825 (H.U.); FONDECYT 3190738 (A.I.S.);

FONDECYT 3180427 (Y.H.), FONDECYT 3150113 (A.C.-S.), and EMBO ASTF 385-2016 (A.C.-S.); CONICYT fellowship (PCHA/Doctorado Nacional/2016-21160232) (M.G.-Q.), MSCA RISE-734749 (INSPIRED) (M.Q.-G.) and FONDAP-GERO-15150012 (A.I.S., P.P.). A.G. is supported by FONDAP-CRG-15090007 and ACT1401. DAA was supported by an Irvington Postdoctoral Fellowship of the Cancer Research Institute. A.A. is supported by FONDECYT 1161065 and AFB170005. G.K. is supported by the Ligue contre le Cancer (équipe labellisée); Agence National de la Recherche (ANR)—Projets blancs; ANR under the frame of E-Rare-2, the ERA-Net for Research on Rare Diseases; Association pour la recherche sur le cancer (ARC); Cancéropôle Ile-de-France; Chancelerie des universités de Paris (Legs Poix), Fondation pour la Recherche Médicale (FRM); a donation by Elior; the European Commission (ArtForce); European Research Area Network on Cardiovascular Diseases (ERA-CVD, MINOTAUR); the European Research Council (ERC); Fondation Carrefour; Institut National du Cancer (INCa); Inserm (HTE); Institut Universitaire de France; LeDucq Foundation; the LabEx Immuno-Oncology; the RHU Torino Lumière; the Seerave Foundation; the SIRIC Stratified Oncology Cell DNA Repair and Tumor Immune Elimination (SOCRATE); the SIRIC Cancer Research and Personalized Medicine (CARPEM); and the Paris Alliance of Cancer Research Institutes (PACRI). P.M.D was supported by FCT LISBOA-01-0145-FEDER-007660, PTDC/NEU-NMC/2459/2014 and IF/00697/2014 (Portugal). R.P. is supported by Fondation pour la Recherche Médicale (FMR, DEQ20180339169) and Institut National de la Santé et de la Recherche Médicale (INSERM). E.C. is supported by Institut National du Cancer (INCa PLBIO), ANR under the frame of ERANET (ERAAT) and EU H2020 MSCA ITN-675448 (TRAINERS) and MSCA RISE-734749 (INSPIRED).

## Author contributions

E.D., M.G-Q., and A.I.S.: conceived and designed the project, carried out experimental work with cell culture and mouse models, and wrote the paper; J.M.B., C.E., H.U., D.S., P.P., A.C.-S., Y.H., E.A.S., D.G., C.V., A.P., D.A.-A., G.C., P.G., A.G.: carried out experimental work and interpreted data; P.G. and E.A.S.: performed bioinformatics work, interpreted data and wrote the manuscript; A.A., P.M.D., Contributed to project design and data interpretation; C.H., P.W., G.K., R.P., E.C.: contributed to project design, interpreted data and wrote the paper.

## Competing interests

The authors declare no competing interests.
