## [Peer Review File · Nature Communications]

Reviewers' comments:

Reviewer #1 (Remarks to the Author):

In this manuscript, Dufey et al. present experimental evidence supporting the novel notion that IRE1a not only operates to govern cellular responses to ER stress, but also to genotoxic stress. Of note, these effects seem to be independent of canonical XBP1 splicing, and are in fact mediated by RIDD. This is a tremendously interesting and relevant story that greatly expands our understanding of IRE1 function beyond the UPR. Of particular importance for the field, the authors present critical evidence demonstrating physiological RIDD in the setting of genotoxic stress. RIDD has been documented mostly in the setting of artificial XBP1 deletion in cell types such as hepatocytes and some dendritic cell populations. Therefore, a key contribution of this study is unearthing that genotoxic stress can evoke IRE1-driven RIDD in the context of XBP1 sufficiency. The experiments are clever, well-performed and rigorous. This is an excellent study by a group of experts in the area of ER stress and IRE1 biology. I only have the following concerns/questions to clarify some parts and further improve the story:

-In Supplementary Fig. 1e, can the authors confirm the observations using Kira8 and MKC8866?

-Fig. 2: The authors claim that IRE1a induces cell cycle arrest. While I understand that IRE1a KO cells are stuck in G0-G1, why is cell proliferation not affected?

-Figs. 2c-e show that Etoposide induces more DNA damage in IRE1a KO cells, and this probably leads to cell cycle arrest (Fig. 2e shows that less IRE1a KO cells enter the S phase). Is this related to the increased apoptosis observed in Fig. 1g?

-In Fig. 2a-b, IRE1a KO cells recover faster from DNA damage and have less pCHK1/2. Why would there be cell cycle arrest and apoptosis in IRE1a KO cells? The authors should explain more clearly the interpretation of these results.

-Fig 3e: Is expression/activity of RtcB (ligase that mediates XBP1 splicing) affected by genotoxic stress? Is this the reason why no XBP1-mediated effects are observed in the system? I appreciate some experiments were done in the setting of Tunicamycin-pretreatment followed by genotoxic stress, but directly evaluating RtcB expression or using additional ER stressors such as TG would be informative and support the claims.

-Fig. 3: Please show quantification for Fig. 3i. Also, Fig. 3j only describes the effects of shPpp2r1a. Does shRuvbl1 revert the effects of IRE1a KO as well? Does shPpp2r1a (and shRuvbl1) in IRE1a KO cells also allow the progression of the cells to the S phase and prevent apoptosis?

-Fig 6c, XBP1 splicing is not evident in Tm-treated samples, although the authors describe it induces mild splicing.

-Finally, is Trp53 differentially expressed/regulated in IRE1-sufficient vs. IRE1-deficient cells undergoing genotoxic stress? Likewise, how is cAbl activated in this setting, and how cAbl-S724 phosphorylation prevents XBP1 splicing by IRE1?

MINOR POINTS:

-Please check the corresponding number of the figures in the text, e.g. page 7 and page 8 line 7.

-In the texts describing Fig. 3d in page 6, line 11, there is a typo “..and Xbpl1 mRNA (Fig.3d).”

-Fig 3e, the bars indicating the treatment and time are not aligned well.

-The texts describing Fig. 4 were mislabeled as Fig 5.

-On page 8, line 7, Fig 5d was mislabeled as Fig.65.

-Page 8, line 25-30, please elaborate the data more clearly, whether the results indicated were obtained from liver or bone marrow. (e.g. In the line 30, ... in this tissue. What does “this tissue” indicate?) Also, can you synchronize the nomenclature of gene p21 and CDKN1 in Fig 6a and c?

-Fig 6 legend, please clarify whether both etoposide or tunicamycin-treated mice were sacrificed at 6h and 16h or each treated group were sacrificed at different time points.

-There are several grammar errors and mistakes in the Reporting Summary File. Please check for accuracy and fix the multiple typos.

Reviewer #2 (Remarks to the Author):

IRE1 α is localized to the endoplasmic reticulum (ER) and plays a central role in regulating the unfolded protein response (UPR) as an endoribonuclease in splicing XBP1 to control expression of UPR genes. A recent report describes a non-canonical role for IRE1 α in recruiting IP3 receptor to mitochondrial associated membrane (MAM) to facilitate Ca²⁺ transfer between the ER and mitochondria. In this report we see evidence from the same group for yet another unexpected role for IRE1 α in the DNA damage response (DDR) to assist in DNA repair and enhance cell survival. This is not too surprising since a connection has already been made between the DDR and proteostasis. Paull and her colleagues (Sci Signal 2018) have provided evidence that ATM, a central player in the DDR, also plays a role in protein homeostasis. As such this report is not entirely novel but it does provide more detailed mechanistic data to explain this connection.

In this study IRE1 α still functions in controlling mRNA, but here specifically in the DDR and show that it is

a conserved function.

Specific comments

1. This is a carefully carried out study that provides good evidence for selective activation of IRE1 α after different types of DNA damage and in its absence the response was impaired. It directly controlled the stability of several mRNAs including Ppp2r1a, coding for the scaffold subunit of PP2A which controls DNA damage kinase activity and Ruvbl1 mRNA involved in chromatin remodelling.
2. While the data in Fig 1 and supplementary Fig 2 point to a role for IRE1 α in cell survival they actually show cell viability data measured with propidium iodide. They should also provide cell survival data, e.g. colony survival.
3. The evidence that IRE1 α functions downstream of ATM activation is good since activation of ATM (pS1981) is unaffected in the absence of IRE1 α (Fig 2f) but the change in ATM protein on the same blot is surprising, considerably increased after IRE1 α KO? Might be instructive to include SMC1 or Kap1 substrates rather than Chk1 if ATM is the kinase?
4. Quality of IRE1 α -c-Abl interaction blot (Fig 4f) poor needs to be more convincing.
5. Reference is made to "possibly other targets" beyond PP2A and pontin. Please elaborate with examples. Why are these mRNAs so critical in RIDD for DDR?

Reviewer #3 (Remarks to the Author):

In this study, Dufey et al. show that IRE1 α is required for DNA damage response and cell fate determination in cultured cells, flies and mice independent of Xbp1 splicing. They further demonstrate that the mRNAs of the DNA repair proteins Ppp2r1a and Ruvbl1 are substrates of IRE1 α -mediated RIDD, which directly connects IRE1 α to the DNA damage response. This work provides a molecular link between genome stability and a key molecule of the proteostasis pathway, and it provides a possible additional explanation for how IRE1 α mutations affect cancer progression. The work is interesting and most of the experiments were well-designed. However, there are a few specific issues that should be addressed prior to publication.

1. Although Xbp1 splicing is the most studied downstream event of IRE1 α signaling, studies have identified other targets of IRE1 α , such as IRE1 α -dependent decay of mRNA (RIDD) and JNK activation. JNK is a key player in cell fate determination. RIDD has also been directly associated with cell death through regulating Death Receptor 5 and caspase 2. All of these targets are very relevant to the topic of this ms. Therefore, besides examination of Xbp1 splicing, these targets should be assessed in response to DNA damage to clarify the relationship between other known pathways and the model proposed in this ms.
2. The mechanism of how IRE1 α is activated by DNA damage is one of the key advances in this ms. Fig.1a shows the increase of IRE1 α protein level upon etoposide treatment, but the mRNA level of IRE1 α did not change (Fig. 6a). Fig. 4f suggests increased interaction between IRE1 α and c-Abl (but is not very convincing, see below). Since c-Abl is a kinase, this implies the involvement of IRE1 α phosphorylation.

However, there is no obvious change of IRE1 α phosphorylation after etoposide treatment in Fig.1a, and the quality of the p-S724-IRE1 α blot in Fig. S5a is not very good? The authors should clarify which mechanism is the dominant/relevant one and expand on this in the discussion.

3. There are many known IRE1 α mutations in cancers. It would be interesting to see their effects by transfecting some of them into the IRE1 α KO cells and examine stability of Ppp2r1a and Ruvbl1 mRNA, and apoptosis. This quick and simple line of experimentation would significantly add to this study.

4. Chk1/2 activity is compromised in IRE1 α KO cells (Fig. 2f, 3j and 6e), without obvious changes in ATM activity. Chk2 is a direct substrate of ATM, and Chk2 can phosphorylate Chk1. According to the close relationship between ATM and Chk1/2, how do the authors explain the ATM-independent regulation of Chk1/2?

5. The data quality of some key data could be improved.

a) Activation of IRE1 α by DNA damage is a very important point of this study. In Fig. 1a, the time point "0" sample was either affected by adjacent lanes or poorly dissolved, and this western blot could be improved. The quality of Fig.S5a should be improved as well. Quantification of these two figures should be included to verify the subtle differences.

b) In Fig. 1g, 2a and 2b, the IRE1 α KO and IRE1 α KO+ IRE1 α -HA were shown for comparison. The wild-type cell control should be included as well.

c) Ppp2r1a and Ruvbl1 are two major downstream targets of IRE1 α proposed by the authors. However, Fig. 3j only shows the effect of Ppp2r1a. In Fig. 4c, Abl shRNA works for both Ppp2r1a and Ruvbl1; while in Fig 4b, only the effect of Abl inhibitor on Ruvbl1 was shown. In Fig. 5b, 6b and 6d, only Ppp2r1a was shown. What is the reason for the selective presentation of these results?

d) In Fig. 4f, the IRE1 α -HA input blot should be shown. The current interpretation that etoposide enhances the interaction between IRE1 α and c-Abl is not convincing, since IRE1 α looks more after etoposide treatment. Perhaps more c-Abl was pulled down by IRE1 α -HA simply because of more IRE1 α , instead of enhanced interaction?

e) The figure numbers do not match with the legend in Fig. S7b and S7. The treatment labeling in Fig. S7c is also not correct. The expression profiling of etoposide and tunicamycin (Fig. S7a and S7b) look very similar to me. It's hard to understand how these similar heatmaps lead to distinct results from GO analyses. I suggest the authors provide more details about this analysis.

UNIVERSITY OF CHILE

Universidad de Chile **Institute of Biomedical Sciences**, Full Professor

THE BUCK INSTITUTE FOR RESEARCH ON AGING

Adjunct Professor

Claudio Hetz, Ph.D.

Genotoxic stress triggers the selective activation of IRE1 α -dependent RNA decay to modulate the DNA damage response

Dufey et al., 2019

Reviewers' Comments

Reviewer #1

Comment 1: *In this manuscript, Dufey et al. present experimental evidence supporting the novel notion that IRE1 α not only operates to govern cellular responses to ER stress, but also to genotoxic stress. Of note, these effects seem to be independent of canonical XBP1 splicing, and are in fact mediated by RIDD. This is a tremendously interesting and relevant story that greatly expands our understanding of IRE1 function beyond the UPR. Of particular importance for the field, the authors present critical evidence demonstrating physiological RIDD in the setting of genotoxic stress. RIDD has been documented mostly in the setting of artificial XBP1 deletion in cell types such as hepatocytes and some dendritic cell populations. Therefore, a key contribution of this study is unearthing that genotoxic stress can evoke IRE1-driven RIDD in the context of XBP1 sufficiency. The experiments are clever, well-performed and rigorous. This is an excellent study by a group of experts in the area of ER stress and IRE1 biology.*

Comment 1: *In Supplementary Fig. 1e, can the authors confirm the observations using Kira8 and MKC8866?*

Answer: We thank this reviewer for the enthusiasm and for highlighting the possible impact of the study to the field, since it will increase our understanding about the role of RIDD under genotoxic stress.

We agree with the reviewer that some of our findings could be strength by a further confirmation using pharmacological inhibitors of IRE1 α , in addition to the genetic strategies already used in this manuscript. To this end, we assessed RIDD under genotoxic stress using the selective IRE1 α RNase inhibitor MKC-8866. In line with our previous results, we observed decrease mRNA levels of the RIDD targets *Bloc1s1* and *Sparc* in MEF wild-type cells treated with Etoposide (Eto) or Tunicamycin (Tm), an effect reversed MKC-8866 treatment, indicating the occurrence of RIDD (**Supplementary Figure 1e and f**). Of note, we decided against the use of KIRA8 or its derivatives as these inhibitors might affect the balance between IRE1 α dimers and oligomers¹ and because of claims about its specificity. Thus, we decided to use specific IRE1 α RNase inhibitors only. These new results support our main findings showing that RIDD activity is selectively engaged under genotoxic stress conditions in the absence of XBP1 mRNA splicing.

Comment 2: *Fig. 2: The authors claim that IRE1a induces cell cycle arrest. While I understand that IRE1a KO cells are stuck in G₀-G₁, why is cell proliferation not affected?*

Answer: We apologize for this misunderstanding. As we show in this manuscript (**Supplementary Figure 3a**), under basal cell culture conditions these cells do not show differences in the rate of proliferation (**Supplementary Figure 3a**) and the cell cycle is normal (**Figure 2e**). We did not test proliferation under DNA damage conditions. The defects on cell cycle control are observed under genotoxic stress. The same results are observed when IRE1 α knockout cells reconstituted with either an empty vector (MOCK) or an HA-tagged full-length IRE1 α are compared (**Supplementary Figure 3b**). Thus, under non-stressed conditions, *IRE1 α does not affect either proliferation or the progression through the cell cycle*. However, under genotoxic stress induced by etoposide, IRE1 α KO cells were arrested in G₀/G₁ phase (**Figure 2e** and **Supplementary Figure 3b**).

Comment 3: *Figs. 2c-e show that Etoposide induces more DNA damage in IRE1a KO cells, and this probably leads to cell cycle arrest (Fig. 2e shows that less IRE1a KO cells enter the S phase). Is this related to the increased apoptosis observed in Fig. 1g?*

Answer: We thank the reviewer for noticing this point regarding cell cycle arrest and cell death. We think that increased DNA damage in IRE1 α KO cells –or more precisely, reduced DNA repair efficiency –leads to G₀/G₁ cell cycle arrest. However, under low levels of genotoxic stress (i.e. short time of exposure to or low doses of etoposide) cell death is not significantly increased, as evidenced by relatively similar levels of Sub-G1 cells (**Figure 2e** and **Supplementary Figure 3b**). Under prolonged or acute exposure to genotoxic stress, IRE1 KO cells become hyper-sensitive to cell death, probably due to unresolved DNA damage caused by an inefficient repair response because of altered signaling regulation (**Figure 1f-g** and **Supplementary Figure 2a-b**). To better clarify the molecular connection between DNA damage and cell death in the revised manuscript, we have evaluated Trp53. Interestingly, we observed an increase in Trp53 phosphorylation in IRE1 α KO cells (**Supplementary Figure 3e**). Thus, we foresee that IRE1 α deficiency leads to alterations in the DNA repair machinery, engaging cell cycle arrest and cell death programs.

Comment 4: *In Fig. 2a-b, IRE1a KO cells recover faster from DNA damage and have less pCHK1/2. Why would there be cell cycle arrest and apoptosis in IRE1a KO cells? The authors should explain more clearly the interpretation of these results.*

Answer: We thanks the comments from the reviewer related with the DNA damage recovery. One of the main conclusions of this work is related to the direct involvement of IRE1 α signaling in mediating the sustained activation of the DNA damage response. We propose that IRE1 α serves as an amplification signal to maintain the response. As requested, we have clarified all these points in the revised discussion of the manuscript and improved the working model for clarity. We argue that IRE1 α KO cells are unable to generate an appropriate response to DNA damage, leading to cell cycle arrest and increased cell death. It is known that the activation of CHK1/2 by DNA damage is necessary for the activation DNA repair programs². In addition, CHK1 can regulate the replication checkpoint and is required for the progression into the S phase and cell

survival³. Indeed, CHK1 knockdown or inhibition leads to reduced proliferation and increased cell death⁴. CHK2 is a kinase downstream of ATM and is particularly important in the response to Double Strand Breaks (DSB), mainly through the phosphorylation and activation of BRCA proteins, promoting DNA repair and the cell cycle progression⁵. Our data shows that IRE1 α do not affect the initial activation of DNA damage sensors, but rather interact through signaling crosstalk with factors that regulate the levels of CHK1/2 and γ -H2AX phosphorylation (i.e. Pontin and PP2A), tuning the response. We have modified the working model for more clarity. We thank this reviewer for these constructive comments.

Thus, increased CHK1/2 inactivation and dephosphorylation of γ -H2AX might not reflect faster recovery but the increased activity of PP2A in IRE1 α KO cells, since *Ppp2r1a* is a newly discovered RIDD target. These changes in PP2A levels can also explain the faster recovery of the signaling events. We have also included additional experiments to demonstrate that RIDD targets mediate the effects in cell cycle and CHK1/2 phosphorylation (**Figures 3i and j; and Fig. S4d, e and f**). We also performed additional cell death assays to confirm our major observations (**Figures 1h and i**).

Comment 5: *Fig 3e: Is expression/activity of RtcB (ligase that mediates XBP1 splicing) affected by genotoxic stress? Is this the reason why no XBP1-mediated effects are observed in the system? I appreciate some experiments were done in the setting of Tunicamycin-pretreatment followed by genotoxic stress, but directly evaluating RtcB expression or using additional ER stressors such as TG would be informative and support the claims.*

Answer: Thanks to the reviewer for proposing this interesting possibility. As requested, we evaluated *RtcB* expression under these conditions. We have included in the new version of the manuscript a western blot that shows the protein expression levels of RtcB under genotoxic stress conditions (**Supplementary Figure 4a**). We observed that RtcB protein levels do not change significantly in MEF WT and IRE1 α KO cells treated with etoposide.

Comment 6: *Fig. 3: Please show quantification for Fig. 3i. Also, Fig. 3j only describes the effects of shPpp2r1a. Does shRuvbl1 revert the effects of IRE1a KO as well? Does shPpp2r1a (and shRuvbl1) in IRE1a KO cells also allow the progression of the cells to the S phase and prevent apoptosis?*

Answer: We agree with the reviewer on this point. Indeed, since IRE1 α directly degrades the mRNAs of *Ppp2r1a* and *Ruvbl1* under genotoxic stress, it is important to assess whether the expression of both proteins in the context of IRE1 α deficiency can rescue the observed phenotypes. To this end, we now added the quantification of the images in the new version of this manuscript (**Figure 3i**) where we show that knockdown of both proteins (shPpp2r1a or shRuvbl1) can rescue the defects on γ -H2AX phosphorylation upon IRE1 α deficiency. In addition, as requested we have performed measurements of cell cycle by FACS in IRE1 α null cells stably transduced with shPpp2r1a or shRuvbl1. Remarkably, the knockdown of both IRE1 α RIDD targets in MEF IRE1 α KO cells were able to recover the percentage of cells that enter into S/G₂/M phase, indicating the exit from G₁/G₀ cell cycle arrest (**Supplementary Figure 4f**). Furthermore, we improved our results on CHK1 phosphorylation by performing indirect immunofluorescence experiments in order to visualize nuclear activation of

CHK1. We observed that the knockdown of either Ppp2r1a or Ruvbl1 in IRE1 α null cells resulted in increased CHK1 phosphorylation in the nucleus compared with controls cells (**Supplementary Figure 4d-e**). These new results support our conclusions about the importance of the RIDD targets Ppp2r1a and Ruvbl1 in the regulation of cell cycle progression under DNA damage and strengthen the molecular link between IRE1 α function and the control of DNA repair.

Comment 7: *Fig 6c, XBP1 splicing is not evident in Tm-treated samples, although the authors describe it induces mild splicing.*

Answer: We have repeated the experiments to detect the mRNA of XBP1s in our bone marrow samples using RT-qPCR. In line with our results, we observed that tunicamycin (Tm) treated mice showed an increase in the levels of XBP1s compared to control groups but we did not observe a clear increase in XBP1 mRNA splicing in mice treated with etoposide, although the same animals show strong responses in the liver (**Supplementary Figure 7c**). Since the effect was very mild, we decided to move this figure to supplementary material mainly because represent an internal control of the experiments. We also included the kidney analyses in this letter as an additional internal control of the Tm treatments performed in the same mice since the MxCre system does not delete in the kidney, serving as an internal control of the Poly(I:C) injection. We hope that this new figures and explanations helped to answer the legitimate concerns of this reviewer regarding this specific issue. We don't have an explanation why levels of XBP1 mRNA splicing versus RIDD change between tissues.

Figure legend: Mouse IRE1 α was conditionally deleted in the liver using the MxCre and LoxP system (IRE1 α cKO) by the Poly(I:C) injection (refer to material and methods section). Mice were intraperitoneally injected with 50 mg/Kg etoposide (Eto) or 1 mg/Kg tunicamycin (Tm) and sacrificed 6 h later. RNA was isolated from kidney of IRE1 α cKO and litter mate control animals, then mRNA levels of XBP1 mRNA splicing was monitored by conventional PCR.

Comment 8: *Finally, is Trp53 differentially expressed/regulated in IRE1-sufficient vs. IRE1-deficient cells undergoing genotoxic stress? Likewise, how is c-Abl activated in this setting, and how c-Abl-S724 phosphorylation prevents XBP1 splicing by IRE1?*

Answer: As we discussed in comment 4, the activation of Trp53 is slightly regulated by IRE1 α under genotoxic stress (**Supplementary Figure 3e**). We argue that the increased phosphorylation and stability of Trp53 in IRE1 α KO cells are a direct consequence of a defective DNA damage response, resulting in more damage and more proapoptotic signals. However, we cannot rule out alternative mechanisms that

regulate Trp53. interestingly, a recent study also suggested that XBP1u, the unspliced version of the protein, controls the stability of Trp53, suggesting an additional connection between the UPR and the DDR⁶. Regarding the concerns about c-Abl-S724 phosphorylation, we apologize if the message was not clear. The phosphorylation at Serine 724 shown in our manuscript corresponds to IRE1 α auto-transphosphorylation residue crucial for IRE1 α activation (this data was removed because of inconsistency). For c-Abl activation we evaluated the tyrosine 412 phosphorylation on c-Abl kinase (**Supplementary Figure 5b**).

Importantly, our results by combining ER stress stimulation with tunicamycin together with etoposide clearly demonstrate that DNA damage do not block XBP1 mRNA splicing. We propose that an alternative activation mechanisms exists that only engages RIDD under DNA damage. This is why we focused in c-Abl since in other settings it can regulate the oligomerization of IRE1 α state toward RIDD¹. In addition, c-ABL is activated downstream of ATM upon DNA damage and that c-Abl is involved in the early stages of the DDR^{7,8}.

We have further improved our data linking the DDR with c-Abl and RIDD using multiple approaches. We generated shRNA cells, in addition to CRISPR KO cells, resulting on a large new figure with complementary evidence supporting our model (**Figure 4**). In order to clarify the putative mechanism of IRE1 α activation by c-Abl we have determined the physical interaction of IRE1 α with c-Abl through co-immunoprecipitation (improved original data, **Figure 4g**) and complemented with a proximity ligation assay (PLA) (**Figure 4h**). In addition, we have performed dimerization/oligomerization studies of IRE1 α using recombinant c-Abl proteins and demonstrated that c-Abl can alone induce the formation of high molecular species of IRE1 α (**Figure 4i**), which are known to correlate with RIDD. These findings support the hypothesis that c-Abl can regulate the oligomerization status of IRE1 α to selectively promote RIDD under genotoxic stress.

MINOR POINTS:

-Please check the corresponding number of the figures in the text, e.g. page 7 and page 8 line 7.

Answer: We thank to the reviewer for noticing these mistakes. We have extensively revised the manuscript correcting this error.

-In the texts describing Fig. 3d in page 6, line 1 1, there is a typo “. and Xbp1 mRNA (Fig.3d).”

-Fig 3e, the bars indicating the treatment and time are not aligned well.

-The texts describing Fig. 4 were mislabeled as Fig 5.

-On page 8, line 7, Fig 5d was mislabeled as Fig.65.

Answer: We thank to the reviewer these comments. We corrected all errors.

-Page 8, line 25-30, please elaborate the data more clearly, whether the results indicated were obtained from liver or bone marrow. (e.g. In the line 30, ... in this tissue.

What does “this tissue” indicate?) Also, can you synchronize the nomenclature of gene *p21* and *CDKN1* in Fig 6a and c? -Fig 6 legend, please clarify whether both etoposide or tunicamycin-treated mice were sacrificed at 6h and 16h or each treated group were sacrificed at different time points.

Answer: We appreciate the comments from the reviewer. In the new version of the manuscript we fixed all this misunderstanding to clarify the specific tissue where we did the measurements.

We thank this reviewer for her/his thorough and comprehensive criticism of our manuscript. We hoped to have answered all her/his concerns in this revised version of the manuscript.

Reviewer #2

IRE1 α is localized to the endoplasmic reticulum (ER) and plays central role in regulating the unfolded protein response (UPR) as an endoribonuclease in splicing XBP1 to control expression of UPR genes. A recent report describes a non-canonical role for IRE1 α in recruiting IP3 receptor to mitochondrial associated membrane (MAM) to facilitate Ca²⁺ transfer between the ER and mitochondria. In this report we see evidence from the same group for yet another unexpected role for IRE1 α in the DNA damage response (DDR) to assist in DNA repair and enhance cell survival. This is not too surprising since a connection has already been made between the DDR and proteostasis. Paull and her colleagues (Sci Signal 2018) have provided evidence that ATM, a central player in the DDR, also plays a role in protein homeostasis. As such this report is not entirely novel but it does provide more detailed mechanistic data to explain this connection. In this study IRE1 α still functions in controlling mRNA, but here specifically in the DDR and show that it is a conserved function.

Specific comments

Comment 1: *This is a carefully carried out study that provides good evidence for selective activation of IRE1 α after different types of DNA damage and in its absence the response was impaired. It directly controlled the stability of several mRNAs including *Ppp2r1a*, coding for the scaffold subunit of PP2A which controls DNA damage kinase activity and *Ruvbl1* mRNA involved in chromatin remodelling. While the data in Fig 1 and supplementary Fig 2 point to a role for IRE1 α in cell survival they actually show cell viability data measured with propidium iodide. They should also provide cell survival data, e.g colony survival.*

Answer: We thank the reviewer for the interest in the study and the highlights about our experimental approaches. We apologize for the low clarity about the novelty of the work. We agree that there were some recent articles suggesting that proteostasis and the DDR are connected however, a possible link with the UPR was not provided. Also, the study mentioned shows that that the machinery involved in the DDR can also impact proteostasis measured as protein aggregation, not ER stress. Our study is suggesting a connection in the other direction where the UPR influences the DDR, which is novel and unexpected.

Regarding the data involving cell viability we agree that the study could be complemented. As requested, we have performed colony assay experiments as readout of survival and recovery in MEF WT and IRE1 α KO cells under genotoxic damage. Unfortunately, we couldn't perform a suitable colony assay in our experimental conditions because our fibroblast when diluted couldn't recover. We spend almost 2 months trying to set up these experiments without success. However, we performed a long termed cell viability experiments using propidium iodide staining in cells pretreated with the IRE1 α RNase inhibitor MKC-8866, where we observed an increase in the cell death in the IRE1 α KO cells (**Figure 1h-i**). This experiment allows us to identify a defect in the capacity to respond and adapt under conditions of genotoxic stress.

Comment 2: *The evidence that IRE1 α functions downstream of ATM activation is good since activation of ATM (pS1981) is unaffected in the absence of IRE1 α (Fig 2f) but the change in ATM protein on the same blot is surprising, considerably increased after IRE1 α KO? Might be instructive to include SMC1 or Kap1 substrates rather than Chk1 if ATM is the kinase?*

Answer: We appreciated the suggestions of the reviewer. We have try several antibodies for total ATM with little success. In the new version of the manuscript we have included the measurement of the two ATM targets suggested, SMC1 and Kap1, as suggested by the reviewer (**Supplementary Figure 3d**). As expected, we found no clear differences in the levels of phosphorylated SMC1 and Kap1 in MEF WT when compared to IRE1 α KO cells, confirming our initial observations. We propose that IRE1 α operates as a signaling modulator downstream of ATM by determining the duration of the DDR through the regulation of the mRNA stability of Ppp2r1a and Ruvbl1.

Comment 3: *Quality of IRE1 α -c-Abl interaction blot (Fig 4f) poor needs to be more convincing.*

Answer: We fully agree with this point. We took these comments seriously and performed extensive experiments to validate and reinforce the connection between c-Abl and IRE1 α . We have improved the quality of the IRE1 α and c-Abl interaction experiments using three complementary approaches. First, we performed co-immunoprecipitation experiments in HEK cells expressing tagged versions of IRE1 α and c-Abl, obtaining better results (**Figure 4g**). Second, in MEF cells, we validated the interaction using Proximity Ligation Assay (Duo-Link) to assess direct interaction between IRE1 α (cells reconstituted with levels equivalent to endogenous IRE1) and endogenous c-Abl (**Figure 4h**). These experiments were also quantified and analyzed followed by statistical analysis. In addition, we showed that this interaction not only occurs at basal levels, but it is also modulated by genotoxic stress (**Supplementary Figure 5h and i**). Finally, we performed *in vitro* oligomerization assays using recombinant cytosolic IRE1 α and full length c-ABL to determine the proportion of IRE1 α monomers and oligomers upon interaction with c-Abl (**Figure 4i**). These experiments also suggest a direct physical interaction between c-Abl and the cytosolic domain of IRE1 α . Using these complementary approaches, we have validated the interaction of IRE1 α and c-Abl and have shown additionally that they not only interact

under DNA damage conditions, but also that c-Abl regulates the oligomeric status of IRE1 α , probably tilting its RNase activity towards RIDD.

In addition, we decided to perform additional experiments to reinforce the data on c-Abl at the cellular level. We used shRNAs, and also generated CRISPR cell lines, to determine the impact of c-Abl on the activation of RIDD under DNA damage. These experiments confirmed the initial observations, and led us to generate a full new figure that defines a possible molecular mechanism linking the DDR with the UPR (**Figure 4**, also see **supplementary figure 5**).

Comment 4: *Reference is made to “possibly other targets” beyond PP2A and pontin. Please elaborate with examples. Why are these mRNAs so critical in RIDD for DDR?*

Answer: We appreciate these comments from the reviewer. We have eliminated this vague statement from the discussion. The degradation of mRNAs by IRE1 α through RIDD is suggested to be cell and context dependent, since the universe of RIDD targets change in different experimental systems⁹⁻¹¹. It seems that the only highly conserved RIDD target is the mRNA of *Bloc1s1*¹² which we have also analyzed. Our idea was to speculate about this idea since we cannot rule out other RIDD targets that participate in the DDR. However, our “reversion” experiments probe that PP2a and RUVBL1 are key to the DDR. The selection of these two genes is solid because it was based on the study of Oikawa et al 2010¹³, where they screened for human RIDD substrates using an unbiased mammalian genome-wide approach for RNAs *in vitro* cleavage, followed by an exon microarray analysis. In this work, they found the cleavage sites for 13 mRNAs.

We would like to thank again this reviewer for all the feedback and also for highlighting the beauty and novelty of the study. This work involved several years of work and the effort of multiple labs to generate a multilayer characterization of a novel biological function for a major UPR stress sensor.

Reviewer #3

In this study, Dufey et al. show that IRE1 α is required for DNA damage response and cell fate determination in cultured cells, flies and mice independent of Xbp1 splicing. They further demonstrate that the mRNAs of the DNA repair proteins Ppp2r1a and Ruvbl1 are substrates of IRE1 α -mediated RIDD, which directly connects IRE1 α to the DNA damage response. This work provides a molecular link between genome stability and a key molecule of the proteostasis pathway, and it provides a possible additional explanation for how IRE1 α mutations affect cancer progression. The work is interesting and most of the experiments were well-designed. However, there are a few specific issues that should be addressed prior to publication.

Comment 1: *Although Xbp1 splicing is the most studied downstream event of IRE1 α signaling, studies have identified other targets of IRE1 α , such as IRE1 α -dependent decay of mRNA (RIDD) and JNK activation. JNK is a key player in cell fate*

determination. RIDD has also been directly associated with cell death through regulating Death Receptor 5 and caspase 2. All of these targets are very relevant to the topic of this ms. Therefore, besides examination of Xbp1 splicing, these targets should be assessed in response to DNA damage to clarify the relationship between other known pathways and the model proposed in this ms.

Answer: We really thank the reviewer for his/her positive and constructive comments of our work. We are also grateful to this reviewer for the great feedback and enthusiasm about our study. We agree that other signaling branches downstream of IRE1 α might affect cell fate such as JNK or the DR5/Caspase-2 signaling. Thus, we have performed additional experiments as requested. In a few papers, it has been proposed that the IRE1 α /TRAF2 interaction triggers ASK1 and JNK pathway leading to apoptosis under ER stress (we believe this idea is overstated, see review by Randy Kaufman 2016 in *Nature*). We measured the levels of JNK phosphorylation in MEF WT and IRE1 α KO cells treated with etoposide. We observed that IRE1 α KO cells show actually an increase in the JNK phosphorylation over time compared to control cells, which might represent increased pro-death signals through the DDR independent of IRE1 α . We decide to exclude this data from the main paper to avoid confusion.

Then we also measured the mRNA levels of DR5 and caspase 2 under genotoxic conditions, since both are regulated by IRE1 α . We noted a mild decrease in caspase 2 mRNA under etoposide treatments in WT cells but not in IRE1 KO cells, consistent with the idea that the mRNA of caspase-2 is indirectly regulated by IRE1 α , as previously reported¹⁴. We did not observe significant changes in the DR5 mRNA expression in these conditions. Since, the caspase-2 and DR5 reports are still controversial due to some studies showing negative data, we respectfully request this reviewer to exclude this data from the paper unless she/he believes this is necessary.

Figure legend: Differential activation of JNK under genotoxic conditions in IRE1 α null cells. WT and IRE1 α KO MEF cells were treated by 0,2,4,8,12 and 16 hours with 10 μ M etoposide and then protein expression of p-JNK/JNK and HSP90 as loading control were measured by western blot.

Figure legend: Caspase 2 and DR5 mRNA expression in WT and IRE1 α KO MEF cells under genotoxic insults. WT and IRE1 α KO MEF cells were treated by 0, 4, 8 and 16 hours with 10 μ M etoposide or Tunicamycin (TM) and then total RNA was extracted. DR5 and caspase 2 mRNA expressions was measured by conventional PCR. XBP1 mRNA splicing was assayed as a control of IRE1 α activation by ER stress.

Comment 2: *The mechanism of how IRE1 α is activated by DNA damage is one of the key advances in this ms. Fig.1a shows the increase of IRE1 α protein level upon etoposide treatment, but the mRNA level of IRE1 α did not change (Fig. 6a). Fig. 4f suggests increased interaction between IRE1 α and c-Abl (but is not very convincing, see below). Since c-Abl is a kinase, this implies the involvement of IRE1 α phosphorylation. However, there is no obvious change of IRE1 α phosphorylation after etoposide treatment in Fig.1a, and the quality of the p-S724-IRE1 α blot in Fig. S5a is not very good? The authors should clarify which mechanism is the dominant/relevant one and expand on this in the discussion.*

Answer: We fully agree with this point. We took these comments seriously and performed extensive experiments to validate and reinforce the connection between c-Abl and IRE1 α . We have improved the quality of the IRE1 α and c-Abl interaction experiments using three complementary approaches. First, we performed co-immunoprecipitation experiments in HEK 293T cells expressing tagged versions of IRE1 α and c-Abl, obtaining better results (**Figure 4g**). Second, in MEF cells, we validated the interaction using Proximity Ligation Assay (Duo-Link) to assess direct interaction between IRE1 α (KO cells reconstituted with IRE1-HA levels equivalent to endogenous) and endogenous c-Abl (**Figure 4h**). These experiments were also quantified and analyzed followed by statistical analysis. In addition, we showed that this interaction not only occurs at basal levels, but it is also modulated by genotoxic stress (**Supplementary Figure 5h and i**). Finally, we performed *in vitro* oligomerization assays using recombinant cytosolic IRE1 α and full-length c-Abl to determine the proportion of IRE1 α monomers and oligomers upon interaction with c-Abl (**Figure 4i**). These experiments also suggest a direct physical interaction between c-Abl and the cytosolic domain of IRE1 α . Using these complementary approaches, we have validated the interaction of IRE1 α and c-Abl and have shown additionally that they not only interact under DNA damage conditions, but c-Abl also regulates the oligomeric status of IRE1 α , probably tilting its RNase activity towards RIDD.

In addition, we decided to perform additional experiments to reinforce the data on c-Abl at the cellular level. We used shRNAs (stable delivery using lentiviruses) and also generated CRISPR cell lines to determine the impact of c-Abl on the activation of RIDD under DNA damage conditions. All this data generated a full new figure that defines a

possible molecular mechanism linking the DDR with the UPR (**Figure 4**, also see **supplementary figure 5**).

In relation to IRE1 α phosphorylation, we postulate that c-Abl binds to IRE1 α and induce its oligomerization to catalyze RIDD. This modulation would not require a direct phosphorylation of IRE1 α by c-Abl (not studied here). The phosphorylation at Serine 724 in IRE1 α ¹⁵ corresponds to a specific auto-phosphorylated residue by the IRE1 kinase domain. To solve this issue we evaluated again the phosphorylation at Serine 724 of IRE1 α under genotoxic conditions; however, we noticed now that a similar signal of phosphoprotein is detected at the same molecular weight of IRE1 α in the IRE1 α KO cells. Thus, we cannot confirm using these antibodies since the antibody has problems with specificity and decided to remove the data. We have contacted the authors to be cautious about the use of this antibody. We include these results here in the response letter for the reviewer consideration. Nevertheless, we improved the phosphorylation results using Phostag experiments to show an electrophoretic shift in the molecular weight of IRE1 α after its phosphoactivation under genotoxic conditions. With these results, we conclude that upon DNA damage c-Abl interacts with IRE1 triggering the formation of dimers and oligomers and its autotransphosphorylation.

Figure legend: Detection of IRE1 α phosphorylation signal in IRE1 α null cells. WT and IRE1 α KO MEF cells were treated 4 hours with 500 mM of Thapsigargin (TG) then IRE1 α total, p-S724-IRE1 α and HSP90 α/β was assayed by western blot. The antibody used for IRE1 α pS724 detection was: Abcam ab48187 rabbit, the antibody for IRE1 α total detection was Cell Signaling CST 3294 rabbit and the antibody used to detect the HSP90 loading control was Santa cruz sc-13119 mouse.

Comment 3: *There are many known IRE1 α mutations in cancers. It would be interesting to see their effects by transfecting some of them into the IRE1 α KO cells and examine stability of Ppp2r1a and Ruvbl1 mRNA, and apoptosis. This quick and simple line of experimentation would significantly add to this study.*

Answer: Thanks to the reviewer for these comments. We agree that several mutations affect *ERN1* in different cancer types and the study of these mutations in the context of our results is very interesting indeed because it may explain the significance of these mutations to the genesis of cancer (genome instability, mutagenesis landscape amplification, etc). We are currently developing a full study on the characterization of the putative effects these mutations may have in the DDR in collaboration with the group of Dr. Chevet in France (Rennes, INSERM). This work also involves the use of animal models of cancer and extensive experimentation and requires at least one year for full completion (one new postdoc and a PhD student are focusing on this project). Since, the work presented here specifically focuses in the mechanisms of IRE1 regulation of the DDR, we estimate that using IRE1 α mutations is out of the scope of our main message. We prefer to wait and present all the data in one complete follow-up study

Comment 4: *Chk1/2 activity is compromised in IRE1 α KO cells (Fig. 2f, 3j and 6e), without obvious changes in ATM activity. Chk2 is a direct substrate of ATM, and Chk2 can phosphorylate Chk1. According to the close relationship between ATM and Chk1/2, how do the authors explain the ATM-independent regulation of Chk1/2?*

Answer: Thanks to the reviewer for the comment. We also agree that our working model was not properly explained in the paper. One of the main conclusions of this work is related to the direct involvement of IRE1 α signaling in mediating the sustained activation of the DNA damage response. We propose that IRE1 α serves as an amplification signal to maintains the response. As requested, we have clarified all these points in the revised discussion of the manuscript. We argue that IRE1 α KO cells are unable to generate an appropriate signaling response to DNA damage, leading to cell cycle arrest and increased cell death. It is known that the activation of CHK1/2 by DNA damage is necessary for the activation DNA repair programs ². In addition, CHK1 can regulate the replication checkpoint and is required for the progression into the S phase and cell survival ³. Our data shows that IRE1 do not affect the activation of DNA damage sensors ATM/ATR, but rather operates through “signaling crosstalk” with factors that regulate the levels of CHK1/2 and γ -H2AX phosphorylation (the RIDD targets Pontin and PP2A), tuning the DDR and the ability to repair and sustain cell viability. We have modified the working model for more clarity and the discussion of the paper for clarity. We thank this reviewed for these constructive comments.

We have also included additional experiments to demonstrate that RIDD targets mediate the effects in cell cycle and CHK1/2 phosphorylation (**Figures 3i and j; and Fig. S4**).

It is also known that CHK2 may be phosphorylated by other kinases involved in the DNA damage response that were not evaluated in this work including DNA-

dependent protein kinase¹⁶ and Polo-Like kinase¹⁷. However, we propose that an alteration in the phosphorylation of the checkpoints kinases by the overexpression/stabilization of PP2A in IRE1 α null cells. This enzyme is the main protein phosphatase responsible for the deactivation of CHK1/2^{18,19}. However, until now many aspects of the inactivation of these proteins remain to be elucidated.

Comment 5: *The data quality of some key data could be improved. a) Activation of IRE1 α by DNA damage is a very important point of this study. In Fig. 1a, the time point "0" sample was either affected by adjacent lanes or poorly dissolved, and this western blot could be improved. The quality of Fig.S5a should be improved as well. Quantification of these two figures should be included to verify the subtle differences.*

Answer: We thank the reviewer for this suggestion. The quality of Figure 1a has been improved in the new version of this manuscript. However, despite multiples attempts of phos-tag assay, we observed always similar results in the qualities of our blots. This is mainly because the antibody against IRE1 α is has background signal and the phos-tag assay is a very difficult technique to perform. For comparison, we include here in the letter a new phos-tag experiment showing similar results.

Figure legend. MEF cells were treated with 10 μ M etoposide (Eto) for indicated time points and then phosphorylation levels of IRE1 α were detected by a Phostag western blot assay (p: phosphorylated 0: non-phosphorylated bands). Total IRE1 α levels were also analyzed using conventional western blot. Cells were treated with 500 ng/mL tunicamycin (Tm) for 8 h as positive control

We have removed Supplementary Figure 5a by reasons exposed in the **comment 2**.

Comment 6: *b) In Fig. 1g, 2a and 2b, the IRE1 α KO and IRE1 α KO+ IRE1 α -HA were shown for comparison. The wild-type cell control should be included as well.*

Answer: We thank the reviewer for this suggestion. We included the wild type cells as a control in some figures mentioned by the reviewer. We believe the best comparison for the reconstituted cells is using Mock controls since the generation of these cells involved the transduction with retroviruses followed by selection with antibiotics. In addition, these cells lines are isogenic, representing the best comparison.

Comment 7: *c) Ppp2r1a and Ruvbl1 are two major downstream targets of IRE1 α proposed by the authors. However, Fig. 3j only shows the effect of Ppp2r1a. In Fig. 4c, Abl shRNA works for both Ppp2r1a and Ruvbl1; while in Fig 4b, only the effect of Abl inhibitor on Ruvbl1 was shown. In Fig. 5b, 6b and 6d, only Ppp2r1a was shown. What is the reason for the selective presentation of these results?*

Answer: Thanks to the reviewer for this comment. The main **Figure 3j** was updated to include the effect of the shRNA against Ruvbl1 in the phosphorylation of CHK11. Regarding Figure 4b we estimated an incomplete inhibition of c-Abl by Imatinib treatments. So, in the new version of the manuscript, we made significant efforts to perform complementary experiments. We used shRNAs to knockdown c-Abl and measure the effects Ruvbl1 and Ppp2r1a, confirming the role of cAbl in the control of RIDD under DNA damage. Furthermore, we generated c-Abl MEF knock out cells using CRISPR/CAS9. Remarkably, these cells demonstrate similar reductions in the RIDD activity on the *Ppp2r1a* and *Ruvbl1* mRNAs, phenocopying the effects observed in IRE1 α KO cells. These results generated a full new figure in the new version of the manuscript (**Figure 4 and Figures S5**).

Comment8. *d) In Fig. 4f, the IRE1 α -HA input blot should be shown. The current interpretation that etoposide enhances the interaction between IRE1 α and c-Abl is not convincing, since IRE1 α looks more after etoposide treatment. Perhaps more c-Abl was pulled down by IRE1 α -HA simply because of more IRE1 α , instead of enhanced interaction?*

Answer: We fully agree that all data regarding the role of c-Abl was weak. As mentioned in the previous point we included we performed additional functional analysis using shRNA cell lines and CRISPR KO cells. For the interaction experiments, we have improved the quality of the IRE1 α -c-Abl interaction, by three complementary techniques. First, we optimized the co-immunoprecipitation assay as you can see in the new main **Figure 4g**. Also, our research group developed a Duo-Link PLA (Proximity ligation Assay, PLA) assay to confirm the interaction message between IRE1 α and c-Abl as you can see in the new main **Figure 4h**. PLA was performed by detecting endogenous c-Abl and using IRE1 α KO reconstituted cells with IRE1 α -HA levels similar to endogenous. Finally, we set up an in vitro oligomerization assay using purified recombinant proteins to demonstrate the changes in the monomers and oligomers proportions of IRE1 α by the physical interaction with c-Abl in the new main **Figure 4i**. These data have now clarified the working model where we provide a possible explanation of how DDR through c-Abl engage IRE1 α oligomerization toward RIDD.

Comment9. *e) The figure numbers do not match with the legend in Fig. S7b and S7. The treatment labeling in Fig. S7c is also not correct. The expression profiling of epotoside and tunicamycin (Fig. S7a and S7b) look very similar to me. It's hard to understand how these similar heatmaps lead to distinct results from GO analyses. I suggest the authors provide more details about this analysis.*

Response: We thanks to the reviewer for the minor's points detected. All these points were solves in the new version of the manuscript.

We would like to thank again this reviewer for the wonderful ideas to clarify the weak points. It took us longer than expected to resubmit a revised version but we wanted to make sure we solved all the points raided by the reviewers.

1. Morita, S. *et al.* Targeting ABL-IRE1 α Signaling Spares ER-Stressed Pancreatic β Cells to Reverse Autoimmune Diabetes. *Cell Metab.* **25**, 883-897.e8 (2017).
2. Bartek, J. & Lukas, J. Chk1 and Chk2 kinases in checkpoint control and cancer. 421–429 (2003).
3. Patil, M., Pabla, N. & Dong, Z. Checkpoint kinase 1 in DNA damage response and cell cycle regulation. *Cellular and Molecular Life Sciences* **70**, 4009–4021 (2013).
4. Syljuasen, R. G. *et al.* Inhibition of Human Chk1 Causes Increased Initiation of DNA Replication, Phosphorylation of ATR Targets, and DNA Breakage. *Mol. Cell. Biol.* **25**, 3553–3562 (2005).
5. Zannini, L., Delia, D. & Buscemi, G. Review CHK 2 kinase in the DNA damage response and beyond. **6**, 442–457 (2014).
6. Huang, C. *et al.* Identification of XBP1-u as a novel regulator of the MDM2/p53 axis using an shRNA library. *Sci. Adv.* (2017). doi:10.1126/sciadv.1701383
7. Shafman, T. *et al.* Interaction between ATM protein and c-Abl in response to DNA damage. *Nature* (1997). doi:10.1038/387520a0
8. Wang, X. *et al.* A positive role for c-Abl in Atm and Atr activation in DNA damage response. *Cell Death Differ.* (2011). doi:10.1038/cdd.2010.106
9. Moore, K. & Hollien, J. Ire1-mediated decay in mammalian cells relies on mRNA sequence, structure, and translational status. *Mol. Biol. Cell* (2015). doi:10.1091/mbc.E15-02-0074
10. Maurel, M., Chevet, E., Tavernier, J. & Gerlo, S. Getting RIDD of RNA: IRE1 in cell fate regulation. *Trends Biochem. Sci.* **39**, 245–254 (2014).
11. Blazanin, N. *et al.* ER stress and distinct outputs of the IRE1 α RNase control proliferation and senescence in response to oncogenic Ras. *Proc. Natl. Acad. Sci.* (2017). doi:10.1073/pnas.1701757114
12. Bright, M. D., Itzhak, D. N., Wardell, C. P., Morgan, G. J. & Davies, F. E. Cleavage of BLOC1S1 mRNA by IRE1 Is Sequence Specific, Temporally Separate from XBP1 Splicing, and Dispensable for Cell Viability under Acute Endoplasmic Reticulum Stress. *Mol. Cell. Biol.* (2015). doi:10.1128/mcb.00013-15
13. Oikawa, D., Tokuda, M., Hosoda, A. & Iwawaki, T. Identification of a consensus element recognized and cleaved by IRE1 alpha. *Nucleic Acids Res.* **38**, 6265–6273 (2010).
14. Sandow, J. J. *et al.* ER stress does not cause upregulation and activation of caspase-2 to initiate apoptosis. *Cell Death Differ.* (2014). doi:10.1038/cdd.2013.168
15. Ali, M. M. U. *et al.* Structure of the Ire1 autophosphorylation complex and implications for the unfolded protein response. *EMBO J.* (2011). doi:10.1038/emboj.2011.18
16. Li, J. & Stern, D. F. Regulation of CHK2 by DNA-dependent protein kinase. *J. Biol. Chem.* (2005). doi:10.1074/jbc.M412445200
17. Bahassi, E. M., Myer, D. L., McKenney, R. J., Hennigan, R. F. & Stambrook, P. J. Priming phosphorylation of Chk2 by polo-like kinase 3 (Plk3) mediates its full activation by ATM and a downstream checkpoint in response to DNA damage. *Mutat. Res. - Fundam. Mol. Mech. Mutagen.* (2006). doi:10.1016/j.mrfmmm.2005.12.002
18. Ryan, C. E. & Piwnicka-worms, H. Phosphorylation of Chk1 by ATR Is Antagonized by a Chk1-Regulated Protein Phosphatase 2A Circuit †. **26**, 7529–7538 (2006).
19. Freeman, A. K., Dapic, V. & Monteiro, A. N. A. Negative regulation of CHK2 activity by protein phosphatase 2A is modulated by DNA damage. *Cell Cycle* (2010). doi:10.4161/cc.9.4.10613

REVIEWERS' COMMENTS:

Reviewer #1 (Remarks to the Author):

The authors have sufficiently addressed most of my comments. I think the manuscript has improved. I would just suggest explaining more deeply the working model in the figure legend (Fig. 6f): clarifying the meaning of the dashed vs full lines and the white arrowhead vs the black ones.

Reviewer #2 (Remarks to the Author):

The authors have addressed all the issues that I raised satisfactorily. While I raised the lack of novelty in connecting the DNA damage response (DDR) to proteostasis they point out that what is novel about their findings is that the unfolded protein response influences the DDR which I accept. It is unfortunate that they could not make a success of the colony forming assays but additional cell viability assays do tend to confirm their initial data . The total ATM determination remains an anomaly but the downstream data for SMC1 and Kap1 are supportive of their findings. The additional data in Fig 4 confirms the IRE1a-c-Abl interaction

Reviewer #3 (Remarks to the Author):

The authors have gone to great lengths to address the criticisms raised during the previous reviews and greatly improved an already important and interesting manuscript. At this stage I strongly support publication in Nature Communications.

UNIVERSITY OF CHILE

Universidad de Chile | Institute of Biomedical Sciences, Full Professor

THE BUCK INSTITUTE FOR RESEARCH ON AGING

Adjunct Professor

Claudio Hetz, Ph.D.

Genotoxic stress triggers the selective activation of IRE1 α -dependent RNA decay to modulate the DNA damage response

Dufey et al., 2020

Reviewers' Comments

Reviewer #1 (Remarks to the Author):

The authors have sufficiently addressed most of my comments. I think the manuscript has improved. I would just suggest explaining more deeply the working model in the figure legend (Fig. 6f): clarifying the meaning of the dashed vs full lines and the white arrowhead vs the black ones.

Answer: We thank to the reviewer for the suggestions to improve our manuscript. To clarify the working model (Figure 6F), we made substantial changes to avoid the dashed lines and white arrowhead. These modifications allow us to simplify the model. Also we improved the explanation of the model in the figure legend following the reviewer suggestion.

Reviewer #2 (Remarks to the Author):

The authors have addressed all the issues that I raised satisfactorily.

While I raised the lack of novelty in connecting the DNA damage response (DDR) to proteostasis they point out that what is novel about their findings is that the unfolded protein response influences the DDR which I accept. It is unfortunate that they could not make a success of the colony forming assays but additional cell viability assays do tend to confirm their initial data. The total ATM determination remains an anomaly but the downstream data for SMC1 and Kap1 are supportive of their findings.

The additional data in Fig 4 confirms the IRE1 α -c-Abl interaction

Reviewer #3 (Remarks to the Author):

The authors have gone to great lengths to address the criticisms raised during the previous reviews and greatly improved an already important and interesting manuscript. At this stage I strongly support publication in Nature Communications

Answer: We would like to thank again the reviewers for their wonderful ideas to clarify the weak points and help us to improve the manuscript.